# High Resolution UDF Meshing via Iterative Networks

**Federico Stella**[1], **Nicolas Talabot**[1], **Hieu Le**[2], **Pascal Fua**[1]

[1]CVLab, EPFL    [2]UNC Charlotte
[1]{firstname.lastname}@epfl.ch, [2]hle40@charlotte.edu

## Abstract

Unsigned Distance Fields (UDFs) are a natural implicit representation for open surfaces but, unlike *Signed* Distance Fields (SDFs), are challenging to triangulate into explicit meshes. This is especially true at high resolutions where neural UDFs exhibit higher noise levels, which makes it hard to capture fine details.

Most current techniques perform within single voxels without reference to their neighborhood, resulting in missing surface and holes where the UDF is ambiguous or noisy. We show that this can be remedied by performing several passes and by reasoning on previously extracted surface elements to incorporate neighborhood information. Our key contribution is an iterative neural network that does this and progressively improves surface recovery within each voxel by spatially propagating information from increasingly distant neighbors. Unlike single-pass methods, our approach integrates newly detected surfaces, distance values, and gradients across multiple iterations, effectively correcting errors and stabilizing extraction in challenging regions. Experiments on diverse 3D models demonstrate that our method produces significantly more accurate and complete meshes than existing approaches, particularly for complex geometries, enabling UDF surface extraction at higher resolutions where traditional methods fail.

## 1 Introduction

Implicit Neural Representations [35, 12, 26] have become powerful tools to accurately model objects whose topology is *a priori* unknown and without being bound to a specific mesh resolution. Those based on Occupancy Fields [23] or Signed Distance Functions (SDFs) [27] are best at handling watertight surfaces. They can be used to represent open surfaces by wrapping a thin volume around them, but this is less than ideal. Unsigned Distance Fields (UDFs) offer a more effective alternative [6, 44, 46, 30, 45, 11, 40] and are used by many surface reconstruction methods [20, 21, 19]. In a typical scenario, a neural network is used to parametrize the field, which is then converted into an explicit mesh for downstream tasks through a meshing process. There are many algorithms to do this conversion well when the field features sign changes to signal surface locations [18, 15, 32, 29]. However, there are no such sign changes in the UDF case and the field is expected to be exactly zero at the surface location.

Most current approaches to meshing UDFs [46, 41, 14, 36] rely on Marching Cubes or Dual Contouring-like algorithms that operate on the vertices of individual voxels to recover the surface that may or may not traverse them. However neural UDFs are notoriously noisy [13, 36, 14], which makes it hard to retrieve the correct surface. Counterintuitively - and contrary to SDFs - increasing the meshing resolution to retrieve finer surface details actually worsens the problem, as shown in the results section (Fig. 5). This is because, at high resolutions, UDF values often become noisier or ambiguous within a voxel, and these local, single-pass methods lack the necessary context to infer the surface reliably, resulting in poorly recovered geometry. As shown in Fig. 1, when the neural UDF fails to reach sufficiently low values, a hole appears in the reconstructed surface. This is

39th Conference on Neural Information Processing Systems (NeurIPS 2025).

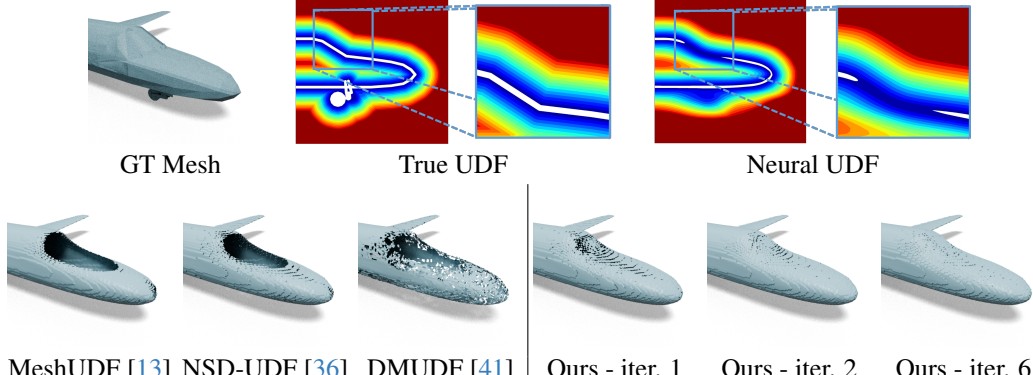

GT Mesh         True UDF        Neural UDF

MeshUDF [13] NSD-UDF [36] DMUDF [41] | Ours - iter. 1   Ours - iter. 2   Ours - iter. 6

Figure 1: **Avoiding holes in a plane cockpit.** (Top) Compared to the ground-truth UDF of a plane, an approximate neural UDF may fail to reach zero, shown in white, at the true surface location. This is clear in the UDF detail shown in the upper right corner, where the two white areas are disconnected. This yields disconnections and holes in the reconstructions, especially at high resolutions. (Bottom) This causes holes in the surfaces reconstructed by current methods. By leveraging neighboring surfaces, our iterative method properly recovers the cockpit nevertheless.

particularly apparent around the planes's cockpit, as shown in the upper right corner of the figure where existing methods do not reliably detect a surface, even though there is a local minimum region. Thus, looking beyond single voxels is necessary, as attempted in [5, 13]. However, these attempts rely on basic propagation heuristics or a simple increase of the receptive fields and still fail at high resolutions, as also shown in the results section.

In this paper, we argue that, to recover the missing parts and to fill the holes, meshing models should incorporate information from previously extracted surface elements, which requires several consecutive extraction passes. To meet this challenge, we introduce an iterative refinement approach. Given a voxel traversed by a surface, it progressively improves the accuracy of the surface recovery by incorporating information previously obtained from increasingly distant voxels. Thus, instead of extracting surfaces in a single pass, we improve the mesh over multiple iterations, where each step integrates newly detected surfaces, distance values, and gradients from neighboring cells.

In extensive experiments on diverse datasets, we show that our spatial propagation helps correcting errors, stabilizing surface extraction, and refining surface localization, particularly where the UDF is ambiguous or noisy, that is, enabling high-resolution meshing where single-pass methods would fail. Thus, our contribution is a novel recursive approach to meshing UDFs that gives much better results than earlier ones at high resolutions.

## 2   Related Work

Signed and unsigned distance fields are now in widespread use to represent 3D surfaces, which are typically parameterized by a latent vector that is decoded by a deep network [23, 27, 6]. Shapes are not tied to a specific resolution, but only to the representational power of the deep network in use. Neural signed distance fields (SDFs) are limited to watertight shapes but are relatively easy to mesh, whereas unsigned distance fields (UDFs) can represent open surfaces. They are used in volume rendering, shape reconstruction from multi-view and shape generation with a variety of techniques [20, 21, 19, 7], but they are much harder to mesh.

**Triangulating Signed Fields.** Marching Cubes (MC) [22, 18] and Dual Contouring (DC) [15] are widely used to mesh signed fields by detecting sign transitions within voxels of a 3D grid. MC interpolates vertex positions along cell edges, while DC places them inside cells and optimizes their locations. Deep-learning adaptations, such as Neural Marching Cubes (NMC) [4] and Neural Dual Contouring (NDC) [5], improve on the classical methods by eliminating the need for manually defined rules or accurate field gradients, respectively. FlexiCubes [32] yields impressive performance on neural SDFs used for shape optimization while McGrids [29] improves the scalability of surface extraction at high resolution. In all methods increasing the resolution, or the number of points used, typically produces more detailed meshes at a higher computational cost. However, as they rely on sign changes, none of these methods is directly applicable to UDFs.

**Triangulating Unsigned Fields.** Meshing neural unsigned distance fields (UDFs) is more difficult than meshing signed ones because the surface is assumed to be where the field *reaches* zero, as opposed to where it *crosses* from positive to negative values. In the latter case, small field value errors may slightly shift the surface location whereas, in the former, it may result in the surface being missed altogether and undesirable holes appearing in the reconstruction, as in Fig. 1.

NDF [6] uses a ball-pivoting algorithm to mitigate this problem. Unfortunately, its reliance on dense point clouds and expensive processing takes hours per mesh and makes it impractical. Methods such as MeshUDF [13] and CAP-UDF [44, 46] assign pseudo-signs to the field values by comparing gradients so that MC can be used, but they are subject to artifacts, especially on complex geometries. MeshUDF partially conditions its pseudo-signs on previously computed voxels by defining a specific voxel-exploration order, but it does so in a single pass and with a heuristic that targets simple shapes such as garments. DCUDF [14] extracts an $\epsilon$-level set using MC, refines it by minimizing a loss function, and cuts it to a single thin surface using a min-cut algorithm [2]. It produces smooth surfaces but requires much manual tuning, long processing times, while sometimes failing to properly cut the inflated surface. While the method refines the vertex locations with an optimization procedure, the mesh extraction is still carried out in a single pass. In NSD-UDF [36], a neural network predicts pseudo-signs starting from local UDF values and gradients, which can be meshed using SDF-like triangulation methods such as MC and DualMesh-UDF [41]. However, all these methods struggle at high resolutions on neural UDFs, where gradients and field values can become very noisy. This interferes with the proper generation of pseudo-signs, surface cuts, or dual vertices, depending on the specific method being used. This can yield inconsistencies and holes in the reconstructed surfaces, which our iterative approach is designed to prevent.

Since DC is a popular alternative to MC, it has also been adapted to work with UDFs. DualMesh-UDF [41] prunes grid cells via a UDF-based threshold and refines vertices using a DC-like quadratic equation. While preserving sharp features, it relies on hand-crafted filtering rules that can produce holes and require manual tuning. This tuning can be partially avoided by using [36] for pre-processing. However the approach still struggles at high resolutions when there is noise. Unsigned Neural Dual Contouring (UNDC) [5] uses a 3D CNN to predict dual vertex locations and connectivity. Training the network requires heavy data augmentation and ignores gradient information, limiting the method's accuracy on neural UDFs [41, 36].

**Iterative Refinement.** Recursive refinement techniques for network predictions have been demonstrated across many domains [24, 28, 37, 33]. These approaches use the surrounding context to improve predictions [31, 9], proving especially effective for tasks such as delineation [34, 10], human pose estimation [25], semantic segmentation [42], and depth estimation [8]. These methods have inspired us to develop the first-ever iterative approach to UDF meshing.

**Mitigating noises in UDFs.** Several approaches have been proposed to learn less noisy UDFs. Zhou et al. [44, 46] enforced parallel level sets near surfaces to improve learning from sparse point clouds. Fainstein et al. [11] use hyperbolic scaling and second-order differentiation to reduce the noise in gradients and ensure the differentiability of the field. Zhao et al. [43] used anchor-based 3D position features to enhance UDF predictions, aiding single-view garment reconstruction. Wang et al. [39] predicted pseudo-signs from surface gradients for meshing, while Venkatesh et al. [38] leveraged surface orientation for better local geometry. However, precise UDF predictions remain challenging, as existing methods struggle to eliminate holes and discontinuities in meshed surfaces.

## 3 Method

A *neural UDF* can typically be written as

$$U_{\mathcal{S}} : \mathbb{R}^3 \times \mathbb{R}^n \to \mathbb{R}^+ \, , \tag{1}$$
$$\mathbf{x}, \mathbf{z} \to d \, ,$$

where $U_{\mathcal{S}}$ is implemented by a deep network. $\mathbf{x}$ is a point in 3D space, and $d$ *should* be the distance from $\mathbf{x}$ to the closest points on the surface $\mathcal{S}$ corresponding to the $n$-dimensional latent vector $\mathbf{z}$ that controls the shape of $\mathcal{S}$. The difficulty with this formulation is that if $U_{\mathcal{S}}$ is not accurate, $U_{\mathcal{S}}(\mathbf{x}, \mathbf{z})$ might not reach zero at places where it should, resulting in unwarranted holes in the surface reconstruction. Since unsigned fields are not differentiable everywhere, they pose a parametrization challenge, resulting in noisy representations [13, 14, 36]. Similarly, the gradients of $U_{\mathcal{S}}$, which are

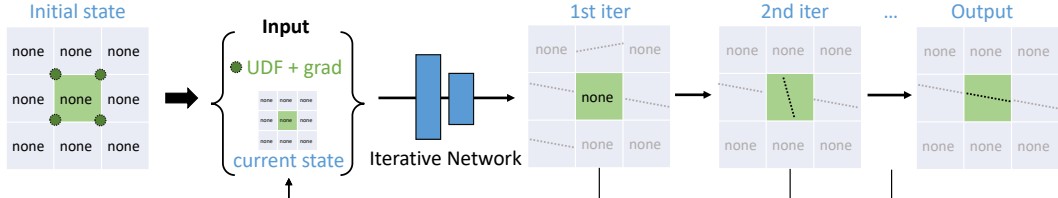

Figure 2: **Iterative Network.** Our model iteratively refines cell configurations. The input consists of the UDF values and gradients at the vertices of the target cell, as well as the current estimated sign configurations of the target and neighboring cells. The surface within a given cell, shown in green, is progressively improved by enforcing consistency with its neighbors.

crucial for meshing [13, 44, 41, 36], can become very noisy when operating at high resolutions, also resulting in reconstruction failures.

Yet, gradients remain accurate further from the surface and there are still reliable cues in parts of the surfaces that are correctly modeled. This should be exploited to overcome the above-mentioned difficulties. To this end, we propose an iterative approach that, at each pass, is conditioned on the surface extracted at the previous pass. By looking beyond single voxels over multiple passes, our method is trained to correct local inconsistencies while preserving the global scene structure. This yields better and more complete surface reconstructions, especially at high resolutions.

### 3.1 From UDF to Triangulated Mesh

At the core of our approach is a per-cell neural network that incorporates neighboring information and refines predictions iteratively. As new information is aggregated, the network progressively improves decisions, reducing noise and enhancing surface extraction for more complete and reliable reconstructions. We first describe its basic architecture and then the iterative refinement process.

**Network Architecture.** Given $U_\mathcal{S}$ of Eq. 1, we query it for UDF values and gradients on a regular grid. We then group the values in cubic cells and feed them to a neural network $f_\theta$, parametrized by $\theta$, whose task is to output a pseudo-sign configuration for the 8 corners of the input cell. As in [36], we use a fully connected feed forward network with $2^7 = 128$ outputs, representing a one-hot encoding of the possible pseudo-sign configurations, up to a complete sign symmetry in the cell.

As shown in Fig. 2, this network takes as input not only the UDF values and gradients at the vertices of the target cell but it also incorporates the current estimated sign configurations of the target and neighboring cells, allowing the model to make new predictions based on its earlier ones, which provides a spatial context. We provide all the architectural details in the supplementary.

**Iterative Refinement.** To refine the predictions and enforce global consistency, the network is run iteratively. This sets up a dynamic process that is trained to reinforce correct predictions, reducing the impact of local noise. Conditioning the network on its previous iterations helps the model retrieve the surface in regions where it previously failed due to noisiness or ambiguity.

More specifically, we set the sign configurations for the first iteration to be zero, which signals the absence of prior information. Note, however, that an all-zero vector is different from the "empty cell" configuration, which is a one-hot vector. This makes the network output predictions based only on the local UDFs and gradient values. In subsequent iterations, we feed the current output back as input. We train the network with a uniformly random number of iterations, with an empirically-determined maximum of 5 additional iterations after the first pass. As a loss function, we use the cross-entropy between the predicted pseudo-signs and the ground-truth signs for all the cells *at every iteration*, making the back-propagation process go through all the iterations.

Formally, the i-th iteration output for $1 \leq i \leq 6$, surface $\mathcal{S}$ and cell $c$ is written as

$$\mathbf{y}_{\mathcal{S},c}^{(i)} = f_\theta \left( U_\mathcal{S}(c), \nabla U_\mathcal{S}(c), \sigma(\mathbf{y}_{\mathcal{S},N_c}^{(i-1)}) \right) \,,$$

$$\mathbf{y}_{\mathcal{S},N_c}^{(i-1)} = \left\|_{c' \in N_c} \mathbf{y}_{\mathcal{S},c'}^{(i-1)} \,,\right.$$

$$\mathbf{y}_{\mathcal{S},c}^{(0)} = [0, 0, ..., 0] \,,$$

(2)

where $\sigma$ is the sigmoid function, $\|$ is the concatenation operator, $N_c$ is the set of cells sharing a face with $c$, including itself, $\mathbf{y}_{\mathcal{S},c}^{(0)}$ is the initial input to the network and $U_{\mathcal{S}}(c)$ is a resolution-normalized UDF, computed by dividing the input UDF by the cell size. For every shape $\mathcal{S}$ in the training dataset and at every epoch, the loss function is taken to be

$$\mathcal{L}_\theta = \sum_{i=1}^{r} \sum_{c \in \mathcal{S}} CE(softmax(\mathbf{y}_{\mathcal{S},c}^{(i)}), GT_{\mathcal{S}}(c)) \tag{3}$$

where $CE$ is the cross entropy, $r$ is a random number between 1 and 6 and $GT_{\mathcal{S}}(c)$ is the ground-truth sign configuration. We observed that, without the sigmoid activation and the randomized number of iterations, the process converges poorly, or even not at all (see supplementary).

To increase robustness to noise, we augment the training values with Gaussian noise. We write

$$U_{\mathcal{S}}(c) \leftarrow U_{\mathcal{S}}(c) * (1 + \mathcal{N}(0, \sigma_{\mathcal{N}})) \, , \tag{4}$$
$$\nabla U_{\mathcal{S}}(c) \leftarrow \nabla U_{\mathcal{S}}(c) * (1 + \mathcal{N}(0, \sigma_{\mathcal{N}})) \, ,$$

where $c$ is a grid cell and $\sigma_{\mathcal{N}}$ is typically set to 1.

**Meshing.** As in NSD-UDF [36], the final output of our pipeline is a pseudo-SDF, that is a UDF in which the grid cell corners have been assigned pseudo-signs so they can be triangulated using existing methods. We use Marching Cubes [18] for its ease of use or DualMesh-UDF [41] for its precision.

## 3.2 Speeding up the Process.

Algorithms such as MeshUDF [13] and NSD-UDF [36] use manually-defined resolution-dependent thresholds that ignore cells that are too far from the surface and, thus, reduce the computational complexity of the algorithm. While this works well at lower resolutions, the high level of noise at higher resolutions makes this process error-prone, resulting in missing regions in the final mesh. Instead, we filter out cells whose vertices have a UDF value equal or above the clamping threshold of the neural UDF, set to be 0.1 in our experiments. In practice, this eliminates 85% of the cells at $256^3$ resolution, with no impact on the final result. For subsequent iterations, since the network outputs a probability distribution over the sign configurations, we can interpret the highest probability as the confidence of the network. We can then conservatively filter out additional cells whose confidence is extremely high ($> 0.999$). As can be seen in Tab. 1a, this substantially reduces the computational cost, making it comparable to that of the other baselines given in Tab. 1b, without accuracy loss compared to meshing all cells.

Table 1: **Filtering and meshing times.** (a) Median L2 Chamfer Distance $\times 10^{-5}$ with 2M sample points (CD), Image Consistency (IC) and model inference times are reported at varying grid resolutions on an auto-decoder [27] trained on ShapeNet [3] cars. The times were measured on an NVIDIA A100 GPU. Notice that our filtering strategies put the proposed pipeline in the same range as existing methods, without impacting its accuracy.

(a) Our filtering strategies speed up the method, maintaining the accuracy within run-to-run variance.

| Res. | Filtering | CD $\downarrow$ | IC $\uparrow$ | Inference time $\downarrow$ |
|---|---|---|---|---|
| | Without filtering | 5.26 | 89.2 | 7m |
| 256 | Low confidence | 5.24 | 89.2 | 1.5m |
| | UDF $< 0.1$ and low confidence | 5.24 | 89.2 | 30s |
| | Without filtering | 8.91 | 88.9 | 1h |
| 512 | Low confidence | 8.87 | 88.9 | 7m |
| | UDF $< 0.1$ and low confidence | 8.88 | 88.9 | 2.5m |

(b) Approximate UDF query and total meshing times at resolution 256. DCUDF makes multiple UDF queries at every iteration, making it difficult to measure the exact query time.

| Time | CAP [46] | MeshUDF [13] | DCUDF [14] | NSD [36] | UNDC [5] | DMUDF [41] | Ours |
|---|---|---|---|---|---|---|---|
| Query | 90s | 30s | / | 30s | 22s | 19s | 30s |
| Total | 3.5m | 35s | 25m | 35s | 30s | 20s | 1m |

# 4 Experiments

## 4.1 Datasets and baselines

We test our method on five different 3D categories, including garments from MGN [1]; cars, chairs, and planes from ShapeNet [3]; and natural scenes from the 3DScene dataset [47], using four different neural UDF architectures. For garments, cars, chairs, and planes, we train a traditional auto-decoder following [27, 36]. To showcase the capabilities of our method on downstream tasks and more precise single-shape neural architectures, we perform surface reconstruction from point clouds by training CAP-L [46][1] on 3D scenes and cars, and DiffUDF [11] on cars. In the supplementary material, we provide additional results using a softplus-based auto-decoder and meshing results obtained from ground-truth UDFs. We use 300 garments and the first 20 samples each for cars, chairs, and planes. As the MGN garments are simpler and exhibit less variety than the other shapes, we use a lower resolution for them. For 3D scenes, we use three scenes from the official CAP-UDF [46] repository[2].

We divide the baselines into two groups, those that rely on Marching Cubes (CAP-UDF [46], MeshUDF [13], DCUDF [14], NSD-UDF [36]) and those that use Dual Contouring (UNDC [5], DualMesh-UDF [41], NSD-UDF [36]). Notably, DCUDF [14] and DualMesh-UDF [41] have been shown to be sensitive to parameter choices. We report results obtained using both the default parameters and those that we manually tuned, denoted as "-T". Due to implementation constraints, DualMesh-UDF and methods that depend on it use a resolution one pixel higher than the others, which we still include in the same tables. UNDC [5] failed at high resolution due to the method's VRAM requirements. DCUDF [14] failed on certain shapes, yielding invalid metrics; these cases are excluded, making its results not directly comparable. Following official recommendations, we also evaluate DCUDF-T-nocut, a variant without the cutting step that produces double-layered meshes, on cars. All meshing methods are run without post-processing to ensure fair comparison.

## 4.2 Metrics

To compare extracted meshes to ground-truth surfaces, we compute three metrics: the L2 Mesh Chamfer Distance (CD) between the meshes, computed with 2M sample points as a two-way point-to-mesh distance; the Image Consistency (IC) [13], which is the product of IoU and the cosine similarity of normal maps rendered from eight viewpoints; and the F1 score [4, 5], also computed with 2M sample points, which measures the portion of the surface within an error threshold of $0.003$ from the ground-truth surface. We report the median of these metrics. It has been found to be more robust to outliers [36] than the mean, which we report in the supplementary for completeness along with the standard deviation.

## 4.3 Comparing against other methods

In Tab. 2a, 2b and Fig. 3, we compare our method against state-of-the-art ones using four auto-decoders on four datasets, spanning from low to high resolutions. Our method consistently outperforms all baselines at high resolutions ($256^3$ and $512^3$) on complex shapes such as cars, chairs, and planes, which are the most prone to suffer from the UDF issues discussed at the beginning of Section 3 and depicted by Fig. 1. This is particularly visible in the three bottom rows of Fig. 3, where our method is the only one able to recover the front of the car, the legs of the chair and the cockpit of the plane. DCUDF [14] yields the smoothest meshes overall, at the cost of losing details and sometimes entire portions of the surface, as shown in Fig. 3, unless the cut operation is avoided, resulting in double surfaces. While previous comparisons [14, 36] have shown CAP-UDF [46] to be often be less accurate than other methods at low resolutions, we notice that the trend is reversed at higher resolutions. There, it retrieves surface portions missing from other baselines. MeshUDF [13] and NSD-UDF [36] can outperform it at high resolutions, but only when their filtering thresholds are removed, as shown in Tab. 4.

At lower resolutions the missing surface problem is less pronounced, and thus iterative refinement does not provide significant advantages. To illustrate this, we present examples of a 3D chair meshed

---

[1]We refer to the learning architecture of CAP-UDF as CAP-L, to distinguish it from the meshing method introduced in the same paper, which we employ as a baseline.

[2]The repository contains five scenes, but "Copyroom" and "Lounge" produced bad metrics across all methods due to inconsistent normalization, so they have been excluded.

Table 2: **Triangulating auto-decoder-based Neural Unsigned Distance Fields.** Median L2 Mesh Chamfer Distance $\times 10^{-5}$ with 2M sample points (CD), F1 score (F1) and Image Consistency (IC) are reported at varying grid resolutions. The best results are in bold. MGN* denotes halved resolution for our MGN experiments due to the lower complexity of the shapes. UNDC failed at resolution 512 due to its large GPU memory requirements.

(a) **Marching Cubes-based methods.**

| Res. | Method | MGN* [1] | | | ShapeNet cars [3] | | | ShapeNet chairs [3] | | | ShapeNet planes [3] | | |
|---|---|---|---|---|---|---|---|---|---|---|---|---|---|
| | | CD ↓ | F1 ↑ | IC ↑ | CD ↓ | F1 ↑ | IC ↑ | CD ↓ | F1 ↑ | IC ↑ | CD ↓ | F1 ↑ | IC ↑ |
| 128 | CAP-UDF [46] | 16.2 | 69.4 | 79.4 | 53.9 | 53.3 | 83.2 | 378 | 50.9 | 70.3 | 14.0 | 69.5 | 80.6 |
| | MeshUDF [13] | 2.45 | 82.9 | 94.1 | 11.3 | 57.7 | 88.6 | 7.43 | 67.6 | 88.0 | 6.82 | 74.6 | 81.0 |
| | DCUDF [14] | 13500 | 2.50 | 2.76 | 6580 | 7.74 | 14.7 | 17000 | 15.9 | 13.9 | 1880 | 27.6 | 24.5 |
| | DCUDF-T [14] | 90.8 | 2.86 | 87.3 | 50.3 | 58.0 | 87.3 | 214 | 64.1 | 82.9 | 144 | 68.8 | 77.2 |
| | DCUDF-T-nocut [14] | - | - | - | 11.4 | **61.7** | **89.4** | - | - | - | - | - | - |
| | NSD-UDF + MC [36] | **1.34** | **83.9** | **94.7** | 6.79 | 59.6 | 88.5 | 6.11 | **67.8** | 88.1 | 3.82 | **79.0** | 84.8 |
| | **Ours** + MC | 2.04 | 81.9 | 94.1 | **5.64** | 59.2 | 88.9 | **3.68** | 66.7 | **88.4** | **3.00** | 78.5 | 84.8 |
| 256 | CAP-UDF [46] | 1.66 | 86.8 | 91.8 | 34.0 | 61.2 | 87.6 | 114 | 70.5 | 82.0 | 5.50 | 83.7 | 85.4 |
| | MeshUDF [13] | 0.958 | 89.7 | 95.0 | 13.6 | 62.3 | 88.6 | 27.8 | **72.9** | 87.3 | 3.47 | 85.8 | 85.2 |
| | DCUDF [14] | 14400 | 4.76 | 3.71 | 346 | 52.6 | 78.2 | 3530 | 49.9 | 54.3 | 27.9 | 84.7 | 82.0 |
| | DCUDF-T [14] | 4.63 | 86.8 | **95.4** | 347 | 52.6 | 78.2 | 3560 | 50.0 | 54.3 | 47.3 | 80.4 | 78.7 |
| | DCUDF-T-nocut [14] | - | - | - | 52.0 | 58.9 | 82.3 | - | - | - | - | - | - |
| | NSD-UDF + MC [36] | **0.808** | **90.0** | 95.2 | 10.2 | 62.0 | 87.9 | 10.9 | 72.3 | 86.2 | 2.91 | 87.3 | 86.0 |
| | **Ours** + MC | 0.878 | 88.9 | 94.9 | **5.23** | **65.0** | **89.2** | **5.14** | 72.9 | **88.8** | **1.84** | **88.7** | **87.0** |
| 512 | CAP-UDF [46] | 0.872 | 90.6 | 94.6 | 31.8 | 61.7 | 87.5 | 63.9 | 71.7 | 82.0 | 5.94 | 87.5 | 86.2 |
| | MeshUDF [13] | 0.798 | 90.6 | **94.8** | 82.7 | 57.0 | 81.7 | 378 | 61.5 | 65.7 | 12.6 | 88.1 | 84.6 |
| | DCUDF [14] | 4.37 | 88.3 | 91.1 | 223 | 56.5 | 84.2 | 2950 | 55.0 | 70.1 | 48.7 | 85.5 | 82.2 |
| | DCUDF-T [14] | 4.38 | 88.2 | 91.1 | 223 | 56.5 | 84.2 | 2000 | 55.0 | 70.1 | 63.0 | 85.4 | 81.3 |
| | DCUDF-T-nocut [14] | - | - | - | 43.1 | 60.4 | 85.6 | - | - | - | - | - | - |
| | NSD-UDF + MC [36] | 0.784 | **90.8** | **94.8** | 56.9 | 58.8 | 83.8 | 295 | 64.7 | 75.7 | 10.0 | 89.4 | 85.1 |
| | **Ours** + MC | **0.722** | 90.6 | **94.8** | **8.84** | **65.6** | **88.9** | **8.76** | **74.5** | **87.2** | **2.37** | **90.9** | **87.1** |

(b) **Dual Contouring-based methods.**

| Res. | Method | MGN* [1] | | | ShapeNet cars [3] | | | ShapeNet chairs [3] | | | ShapeNet planes [3] | | |
|---|---|---|---|---|---|---|---|---|---|---|---|---|---|
| | | CD ↓ | F1 ↑ | IC ↑ | CD ↓ | F1 ↑ | IC ↑ | CD ↓ | F1 ↑ | IC ↑ | CD ↓ | F1 ↑ | IC ↑ |
| 128 | UNDC [5] | 1.09 | 87.1 | 94.1 | 13.5 | 61.7 | 86.4 | 29.9 | 69.4 | 81.9 | 2.50 | 82.0 | 86.1 |
| | DualMesh-UDF [41] | 216 | 68.1 | 68.4 | 952 | 34.4 | 45.5 | 7930 | 12.1 | 9.08 | 112 | 74.3 | 76.1 |
| | DualMesh-UDF-T [41] | 0.806 | 89.9 | 95.4 | 5.56 | 63.5 | 89.5 | 5.34 | **75.1** | 89.1 | 1.96 | 84.3 | 87.5 |
| | NSD-UDF + DualMesh-UDF [36] | **0.760** | 90.4 | **95.0** | 6.34 | 65.7 | 89.1 | 5.50 | 72.8 | 89.2 | 2.08 | 87.1 | 86.6 |
| | **Ours** + DualMesh-UDF | 0.787 | **90.5** | 94.9 | **4.80** | **66.2** | **89.7** | **3.39** | 72.8 | **89.8** | **1.56** | **87.7** | **88.2** |
| 256 | UNDC [5] | 0.931 | 89.1 | 91.5 | 82.4 | 52.3 | 71.4 | 293 | 54.2 | 57.8 | 11.6 | 83.6 | 80.7 |
| | DualMesh-UDF [41] | 176 | 66.3 | 66.4 | 846 | 34.0 | 45.1 | 8280 | 12.4 | 9.26 | 105 | 77.9 | 76.0 |
| | DualMesh-UDF-T [41] | 0.722 | **91.2** | **95.1** | 10.6 | 64.3 | 87.0 | 22.8 | **72.2** | 84.1 | 2.43 | 88.2 | 87.0 |
| | NSD-UDF + DualMesh-UDF [36] | 0.671 | 91.0 | 94.6 | 10.5 | 64.9 | 87.2 | 14.6 | 70.7 | 84.4 | 2.78 | 88.9 | 85.6 |
| | **Ours** + DualMesh-UDF | **0.662** | 91.2 | 94.7 | **5.48** | **65.7** | **88.8** | **4.97** | 71.9 | **86.4** | **1.87** | **90.0** | **87.5** |
| 512 | UNDC [5] | 2.39 | 84.8 | 82.8 | - | - | - | - | - | - | - | - | - |
| | DualMesh-UDF [41] | 167 | 63.7 | 63.9 | 870 | 32.5 | 43.0 | 8190 | 11.8 | 9.07 | 111 | 77.6 | 74.2 |
| | DualMesh-UDF-T [41] | 0.827 | **90.2** | **93.2** | 37.8 | 58.2 | 79.3 | 72.7 | 63.7 | 72.5 | 4.77 | 89.0 | 84.0 |
| | NSD-UDF + DualMesh-UDF [36] | 0.787 | 89.7 | 92.5 | 60.5 | 57.5 | 79.9 | 296 | 62.5 | 68.0 | 9.95 | 87.6 | 83.3 |
| | **Ours** + DualMesh-UDF | **0.726** | 89.7 | 92.8 | **9.65** | **63.0** | **85.6** | **10.1** | **70.4** | **81.9** | **2.51** | **90.1** | **85.8** |

at resolutions of $128^3$, $256^3$, and $512^3$ in Fig. 5, comparing one of the most accurate methods, NSD-UDF [36], with our approach. As shown, at lower resolutions NSD-UDF reconstructs good surfaces, leaving less room for improvement, however it fails to recover many surface regions at $256^3$ and especially at $512^3$ resolution, whereas our method handles these cases more effectively. Similarly, our method does not show significant advantages on the MGN dataset, where the shapes are less complex and nearly all methods achieve high accuracy without missing large portions of the surface.

We also observe that, while the pseudo-signs of NSD-UDF partially mitigate the need for tuning DualMesh-UDF when used in conjunction with it, our pseudo-signs are more robust and consistently outperform both NSD-UDF and the tuned DualMesh-UDF, as shown in Tab. 2b.

Comparing primal and dual methods, we notice that dual ones tend to be more accurate at lower resolutions, while primal ones have an edge at higher resolutions. Note that, while achieving very good CD scores on MGN at high resolution, dual methods tend to create many holes in the reconstructed meshes, as can be seen in the top row of Fig. 3. This does not impact the Chamfer Distance much due to the small size of these holes, but it affects the F1 and IC scores that are better in MC-based methods. Notice also that most primal methods show significant blockiness artifacts at high resolution, including our pipeline paired with Marching Cubes, a common issue with currently

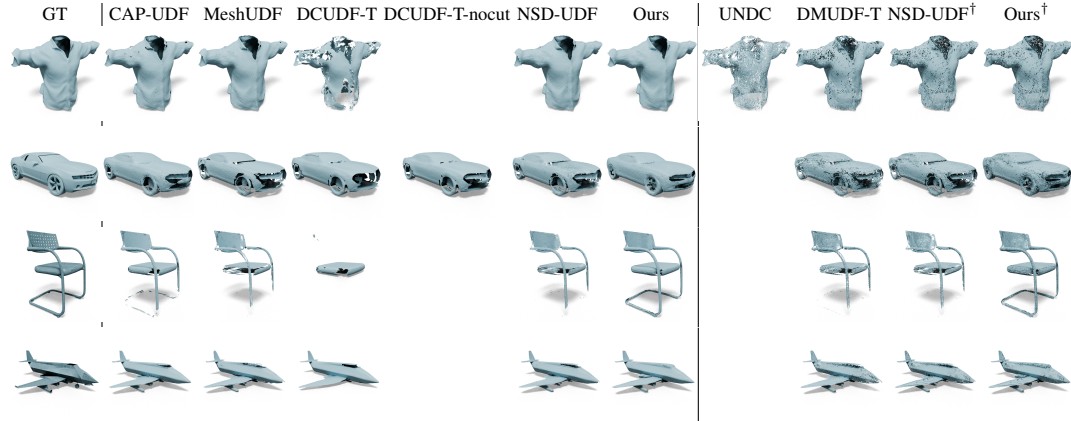

Figure 3: **Qualitative comparison with existing methods (auto-decoders).** Surface meshing results of neural UDFs with all methods at resolution of 512 (and 256 for MGN). UNDC failed at resolution 512 due to high GPU memory requirements. [†] indicates that the method is combined with DMUDF.

Table 3: **Neural Unsigned Distance Fields from point clouds.** L2 Mesh Chamfer Distance $\times 10^{-5}$ with 2M sample points (CD), F1 score (F1) and Image Consistency (IC) are reported at varying grid resolutions. Median scores are reported for cars; mean for scenes due to the low number of samples. The best results are in bold.

| Res. | Method | CAP-L scenes [46] | | | CAP-L cars [46] | | | DiffUDF cars [11] | | |
|---|---|---|---|---|---|---|---|---|---|---|
| | | CD↓ | F1↑ | IC↑ | CD↓ | F1↑ | IC↑ | CD↓ | F1↑ | IC↑ |
| 128 | CAP-UDF [46] | 4.27 | 68.0 | 83.8 | 10.7 | 44.5 | 85.8 | 6.71 | 58.0 | 86.3 |
| | MeshUDF [13] | 4.40 | 68.2 | 84.5 | 9.83 | 45.4 | 85.7 | 9.02 | 59.7 | 86.7 |
| | DCUDF-T [14] | 279 | 59.7 | 84.0 | 45.5 | 41.4 | 85.5 | 32.7 | 49.6 | 89.6 |
| | DCUDF-T-nocut [14] | - | - | - | 8.47 | 47.2 | **88.3** | 12.5 | 56.9 | **90.2** |
| | NSD-UDF + MC [36] | **3.34** | **69.8** | **86.7** | 8.14 | 47.1 | 86.4 | **4.18** | **65.7** | 87.7 |
| | **Ours** + MC | 3.44 | 69.3 | 85.3 | **7.03** | **48.3** | 86.8 | 5.46 | 62.3 | 87.4 |
| 256 | CAP-UDF [46] | 3.65 | 70.1 | 86.1 | 7.51 | 48.0 | 86.6 | 3.72 | 67.9 | 86.7 |
| | MeshUDF [13] | 3.42 | 69.6 | 86.3 | 8.65 | 47.5 | 86.1 | 5.18 | 65.9 | 87.2 |
| | DCUDF-T [14] | 4.75 | 69.9 | 84.6 | 31.1 | 45.5 | 85.7 | 25.5 | 58.3 | 86.4 |
| | DCUDF-T-nocut [14] | - | - | - | 9.58 | 47.9 | 86.0 | 5.56 | 69.5 | 87.7 |
| | NSD-UDF + MC [36] | 3.30 | 70.3 | 86.3 | 8.76 | 47.9 | 86.3 | 3.63 | **70.0** | 86.9 |
| | **Ours** + MC | **3.07** | **71.4** | **86.6** | **7.22** | **50.1** | **87.1** | **3.45** | 68.9 | **88.3** |
| 512 | CAP-UDF [46] | 3.57 | 70.5 | 86.1 | 8.11 | 48.1 | 86.4 | 4.60 | 67.1 | 84.4 |
| | MeshUDF [13] | 4.49 | 69.9 | 84.5 | 8.46 | 47.6 | 85.9 | 3.83 | 67.4 | 86.4 |
| | DCUDF-T [14] | 140 | 68.9 | 83.3 | 31.5 | 45.4 | 85.8 | 17.2 | 60.1 | 84.3 |
| | DCUDF-T-nocut [14] | - | - | - | 11.8 | 46.3 | 84.8 | 5.53 | 69.0 | 87.1 |
| | NSD-UDF + MC [36] | 3.72 | 70.0 | 84.0 | 9.85 | 47.3 | 85.9 | 5.63 | 66.0 | 82.0 |
| | **Ours** + MC | **3.08** | **71.8** | **86.8** | **7.82** | **49.8** | **86.6** | **3.03** | **71.7** | **88.1** |

known architectures for neural UDFs. DCUDF and dual methods reduce this problem thanks to their optimization procedures, but they show more missing surfaces compared to primal methods. Smoothing and mesh-repair algorithms can be used nevertheless to mitigate this in all methods, but they are not applied here to better evaluate the direct output of each method.

In Tab. 3 and Fig. 4, we show additional experiments using two single-shape neural UDF architectures trained on point clouds: CAP-L [46] and DiffUDF [11]. For 3D scenes we used the $100\ points/m^2$ setting from the CAP-UDF [46] paper, using the provided data. For cars, we sample 10k noise-free points per shape following the CAP-UDF protocol. Both architectures are trained using default parameters from their respective repositories. The results follow similar trends as in the auto-decoder experiments: our method achieves higher accuracy than the baselines, especially at high resolutions. Additional DC-based results and experimental details are included in the supplementary material.

## 4.4 Importance of Correct Thresholding

As described in Sec. 3.2, a correct filtering strategy is crucial to keeping the computational costs under control when meshing neural UDFs at high resolution. MeshUDF [13] and NSD-UDF [36] rely on filtering out high UDF values to speed up the computation, which can be far from lossless at high resolutions. Removing such thresholds can help the algorithms to better retrieve surfaces at high

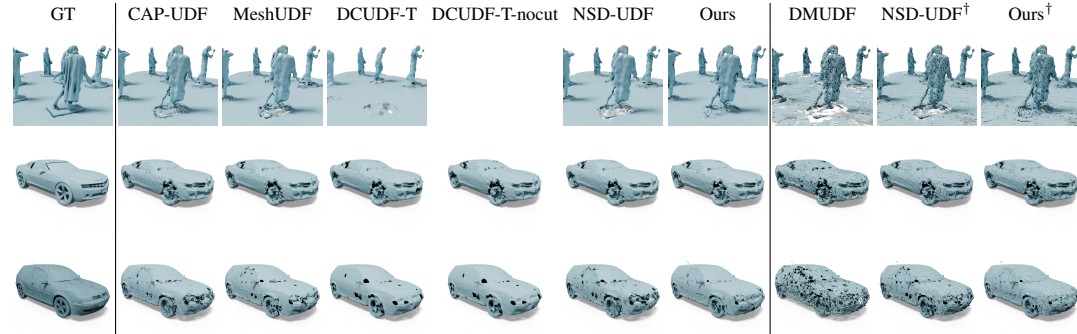

Figure 4: **Qualitative comparison with existing methods (reconstruction from point clouds).** Surface meshing results of neural UDFs at resolution of 512. Top: CAP-L 3D scenes. Middle: CAP-L car. Bottom: DiffUDF cars. † indicates that the method is combined with DMUDF.

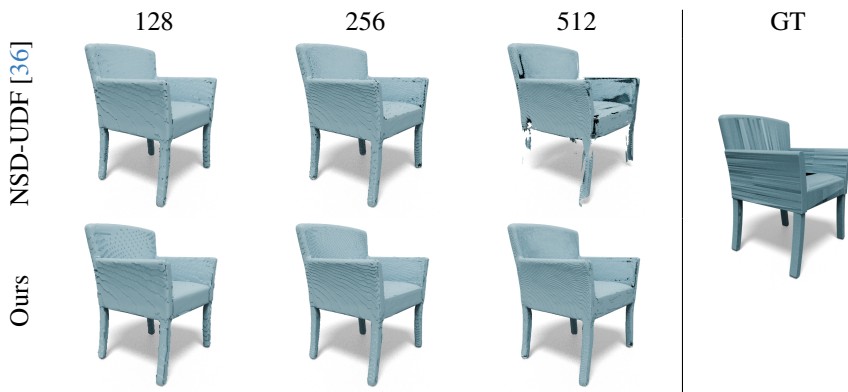

Figure 5: **Meshing at different resolutions.** While NSD-UDF [36] retrieves most of the surface well at a low resolution, it struggles at higher ones. In contrast, our method, recovers the surface well at all resolutions. We use Marching Cubes with both methods.

Table 4: **Removing thresholds on existing methods.** Median L2 Chamfer Distance $\times 10^{-5}$ with 2M sample points (CD ↓) at varying grid resolutions, best results in bold. †Thresholds removed. *Resolution is halved.

| Res. | Method | MGN* | Cars | Chairs | Planes | Res. | Method | MGN* | Cars | Chairs | Planes |
|------|--------|------|------|--------|--------|------|--------|------|------|--------|--------|
| 256 | MeshUDF | 0.958 | 13.6 | 27.8 | 3.47 | 512 | MeshUDF | 0.798 | 82.7 | 378 | 12.6 |
| | MeshUDF† | 1.97 | 8.01 | 6.41 | 3.73 | | MeshUDF† | 0.919 | 9.66 | 20.1 | 2.73 |
| | NSD+MC | **0.808** | 10.2 | 10.9 | 2.91 | | NSD+MC | 0.784 | 56.9 | 295 | 10.0 |
| | NSD+MC† | **0.808** | 8.87 | 6.54 | 2.86 | | NSD+MC† | 0.766 | 23.3 | 28.9 | 4.49 |
| | Ours+MC | 0.878 | **5.23** | **5.14** | **1.84** | | Ours+MC | **0.722** | **8.84** | **8.76** | **2.37** |

resolution, but this is insufficient by itself. Tab. 4, computed on the auto-decoder experiments, shows that there is a noticeable improvement at high resolutions, but the results are still not as good as those of our iterative pipeline. MeshUDF can become less accurate at resolution 256 due to artifacts introduced in regions of the space with high UDF values, which its heuristic is not designed to handle. In contrast, our method is robust to all such perturbations.

Table 5: **Mesh refinement over iterations.** Median L2 Chamfer Distance $\times 10^{-5}$ with 2M sample points (CD ↓) at varying grid resolutions, showing increasing precision over iterations in most cases. *Resolution is halved.

| Dataset | 128 | | | | | | 256 | | | | | | 512 | | | | | |
|---------|-------|-------|-------|-------|-------|-------|-------|-------|-------|-------|-------|-------|-------|-------|-------|-------|-------|-------|
| | It. 1 | It. 2 | It. 3 | It. 4 | It. 5 | It. 6 | It. 1 | It. 2 | It. 3 | It. 4 | It. 5 | It. 6 | It. 1 | It. 2 | It. 3 | It. 4 | It. 5 | It. 6 |
| MGN* | 1.46 | 1.53 | 1.73 | 1.88 | 1.98 | 2.04 | 0.814 | 0.811 | 0.834 | 0.857 | 0.869 | 0.878 | 0.748 | 0.730 | 0.716 | 0.720 | 0.719 | 0.720 |
| Cars | 7.47 | 6.85 | 6.06 | 5.68 | 5.65 | 5.65 | 6.69 | 6.28 | 5.69 | 5.44 | 5.31 | 5.24 | 11.2 | 10.2 | 9.63 | 9.18 | 8.98 | 8.88 |
| Chairs | 4.91 | 4.43 | 3.88 | 3.78 | 3.77 | 3.66 | 5.56 | 5.48 | 5.32 | 5.19 | 5.15 | 5.15 | 10.9 | 10.4 | 9.55 | 9.43 | 8.97 | 8.87 |
| Planes | 3.69 | 3.47 | 3.10 | 3.08 | 3.03 | 3.00 | 2.70 | 2.28 | 2.11 | 2.04 | 1.88 | 1.85 | 3.43 | 2.87 | 2.64 | 2.47 | 2.40 | 2.38 |

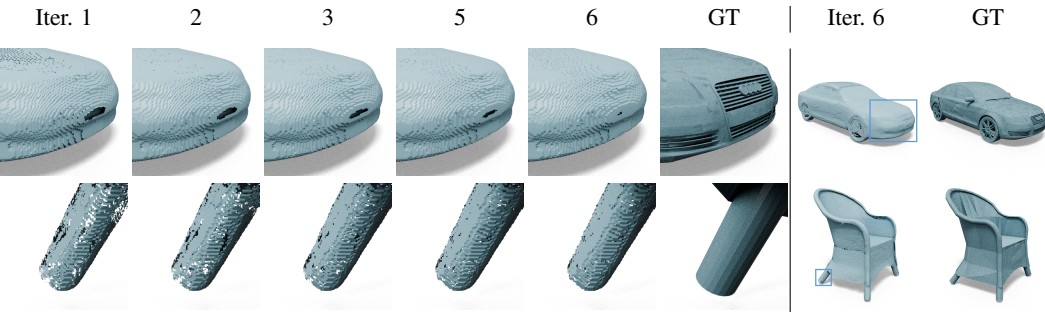

Figure 6: **Meshing iterations.** Intermediate results over the iterations. Propagating information across cells helps fill gaps and retrieve missing surfaces. See the car front and the chair leg. We show the fully reconstructed shapes on the right.

## 4.5   Mesh Evolution over Successive iterations

In Tab. 5 and Fig. 6, we show the evolution of the meshes over several iterations of our scheme on auto-decoder-based UDFs. The meshes are progressively refined and the missing regions filled, with the Chamfer Distance decreasing in most cases, especially at higher resolutions and in complicated shapes such as cars, chairs and planes. The only partial exception to this trend occurs in the MGN dataset, where the mesh is already very accurate after the first iteration and remains so during the following ones at high resolutions. At a lower resolution, such as 64, we see the opposite behavior, especially for simple shapes like garments. The problems described in Fig. 1 do not arise, lessening the usefulness of an iterative scheme. We also observe that our pipeline often produces more accurate meshes than the baselines already during the first iteration, as shown in Fig. 1 and by comparing results in Tab. 5 and 4. This is mainly due to the more robust training of our network compared to NSD-UDF [36] and thanks to the asbsence of heavy thresholds during meshing.

## 5   Limitations

Due to its iterative nature, our pipeline is naturally slower than single-pass or heavily thresholded approaches. However, the system's speed can be improved by reducing the number of iterations, which comes with a tradeoff in accuracy as shown in Tab. 5, or by filtering cells more aggressively. Furthermore, as shown in Tab. 1b, querying input UDFs and gradients often constitutes a significant portion of inference time, an operation performed only once in our iterative pipeline. As a result, the additional iterations in our approach have a comparatively small impact on the overall computation time, which remains in the ballpark as those of the fastest baselines.

For simpler shapes, such as garments, methods specifically designed to prioritize topology over accuracy, such as MeshUDF, may sometimes be more suitable. Moreover, while allowing high resolution surface extraction, our method can still produce artifacts in the form of irregular boundaries and small holes, as visible in Figs. 3 and 6. Post-processing algorithms can help with this, but future work should focus on improving in these aspects.

## 6   Conclusion

Our iterative approach to meshing neural UDFs achieves state-of-the-art accuracy and robustness, enabling surface extraction for complex shapes at high resolutions. By refining surfaces over multiple iterations, it addresses the challenges posed by noisy distance fields, which can create severe problems for heuristic-based methods. We believe this represents a paradigm shift in using neural surface localization to capture the complex interplay between neural distance fields and reconstructed surfaces—far from a trivial task.

Looking ahead, to improve high-resolution surface extraction on UDFs key directions include optimizing computational efficiency, exploring adaptive iteration strategies, and improving topology preservation. End-to-end learning for unsigned distance fields and meshing remains a critical frontier, bringing us closer to seamless, high-fidelity 3D reconstruction across diverse applications.

# 7 Acknowledgements

This work was funded in part by the Swiss National Science Foundation.

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

## A.1 Complete quantitative results

For the sake of completeness, we report in Tabs. A.3a, A.3b, A.4a, A.4b, A.5a, A.5b, A.6a and A.6b the mean results on the datasets and methods used in the main paper, alongside their respective standard deviations. We also report the results of DC-based methods on point cloud reconstruction from CAP-L [46] and DiffUDF [11] in Tab. A.2. We observe that the trends are consistent with the median results shown in the main paper, with our method outperforming all baselines at high resolutions on complex shapes, and competing closely with existing methods at lower resolutions and on simpler shapes. The standard deviations computed on our method are also consistently lower than the baselines at high resolution, while remaining competitive at lower resolutions, showing that our method is also more robust to shape variations compared to existing baselines. As in the main tables, notice that UNDC [5] failed to reconstruct meshes at high resolution due to the method's VRAM requirements. DCUDF [14] failed to reconstruct some of the shapes in the experiments, producing unbound metrics. We discard such shapes from the reported results of DCUDF, which means that its numbers are not directly comparable to the other baselines.

## A.2 Implementation details and other ablation studies

Table A.1: **Number of additional training iterations.** Median Image Consistency (IC $\uparrow$) of the *last* iteration at resolution 512. *Resolution is halved.

| Max training iter. | MGN* | Cars | Chairs | Planes |
|---|---|---|---|---|
| 1 iter. | 94.9 | 88.5 | 86.2 | 87.0 |
| 3 iter. | 94.9 | 88.4 | 86.0 | 86.8 |
| 5 iter. | 94.8 | 88.9 | 87.2 | 87.1 |
| 7 iter. | 94.9 | 88.5 | 85.9 | 86.5 |

Our network architecture consists of 2 fully connected hidden layers, with 1024 nodes each, and an output layer with 128 outputs. The input layer accepts UDF values and gradients at the 8 cell corners, making up 32 inputs. Additionally, 128 inputs per cell are needed to enable multiple iterations. We consider the current cell and the 6 cells that share a face with it, for a total of $7 * 128$ additional inputs, which brings the total number of input nodes to 928 and the total number of trainable weights to around 2.1M. Each layer, except for the final one, is followed by a leaky ReLU activation function with a negative slope of 0.01. The output layer is followed by a sigmoid activation before being used as input for the next iteration, and by a softmax function for the cross entropy loss. The network is trained using the Adam optimizer [16] with a learning rate of $5 \times 10^{-4}$ for 50 epochs.

The learning rate is lower than in [36], with more epochs. We have found this helps our iterative process converge better. For training, we use the first 80 watertight shapes from ABC [17] sampled at resolution $128^3$, yielding around 5.5M training cells. As mentioned in the method section of the main paper, we limit the number of additional training iterations to 5. In Tab. A.1, we provide an ablation study showing that training the network with a single additional pass already achieves good results, with 5 iterations achieving the best. Using even more iterations did not bring measurable benefits. The training takes around 2 hours on an NVIDIA A100-40G GPU.

As an ablation, we also trained the network without the noise augmentation described in the method section of the main paper. The method presented in this work achieved a median Mesh Chamfer Distance $\times 10^{-5}$ of 5.64, 5.23 and 8.84 at resolutions $128^3$, $256^3$ and $512^3$ respectively, using the UDF auto-decoder trained on ShapeNet [3] cars in Section 4.3 of the main paper. Without noise augmentation, the same experiment achieved 8.01, 12.5 and 47.8 respectively, showing that the noise augmentation is crucial to achieve good performance in practical scenarios.

## A.3 Training convergence

In the method section of the main paper we state that our pipeline applies a sigmoid function to the network outputs before using them as input for the next iteration. We have experimentally found that, when using an identity activation (i.e. direct input) and training with only one iteration, the network goes from an IC of 87.0 on ShapeNet [3] cars at resolution 512 to 88.6, however it starts diverging

Table A.2: **Neural Unsigned Distance Fields from point clouds (DC-based methods).** L2 Mesh Chamfer Distance $\times 10^{-5}$ with 2M sample points (CD), F1 score (F1) and Image Consistency (IC) are reported at varying grid resolutions. Median scores are reported for cars; mean for scenes due to the low number of samples. The best results are in bold. UNDC failed at resolution 512 due to its large GPU memory requirements.

| Res. | Method | CAP-L scenes [46] | | | CAP-L cars [46] | | | DiffUDF cars [11] | | |
|---|---|---|---|---|---|---|---|---|---|---|
| | | CD ↓ | F1 ↑ | IC ↑ | CD ↓ | F1 ↑ | IC ↑ | CD ↓ | F1 ↑ | IC ↑ |
| 128 | UNDC [5] | **3.34** | 69.7 | 84.4 | 7.85 | 49.4 | 85.5 | 4.46 | 59.1 | 87.6 |
| | DualMesh-UDF [41] | 53.5 | 63.4 | 69.5 | 9.44 | 50.4 | 85.6 | 6.03 | 62.3 | 84.9 |
| | NSD-UDF + DualMesh-UDF [36] | 33.6 | 69.8 | **84.5** | 5.49 | 52.4 | 87.2 | **3.07** | **68.2** | **88.2** |
| | **Ours** + DualMesh-UDF | 33.6 | **70.0** | 84.4 | **5.35** | **52.8** | **87.3** | 3.68 | 65.3 | 88.1 |
| 256 | UNDC [5] | **7.85** | **69.1** | 76.9 | 7.77 | 48.5 | 84.2 | 6.84 | 56.1 | 80.9 |
| | DualMesh-UDF [41] | 50.4 | 63.0 | 68.3 | 8.51 | 49.2 | 84.9 | 5.01 | 61.9 | 83.1 |
| | NSD-UDF + DualMesh-UDF [36] | 34.3 | 68.9 | **83.4** | 6.74 | 49.7 | 86.5 | **2.72** | **72.3** | 88.3 |
| | **Ours** + DualMesh-UDF | 220 | 49.9 | 79.6 | **6.35** | **50.6** | **86.9** | 2.89 | 70.0 | **88.5** |
| 512 | UNDC [5] | - | - | - | - | - | - | - | - | - |
| | DualMesh-UDF [41] | 51.7 | 59.8 | 61.8 | 9.26 | 46.3 | 83.5 | 5.07 | 59.4 | 79.4 |
| | NSD-UDF + DualMesh-UDF [36] | 35.2 | 66.5 | 77.9 | **7.69** | 47.0 | 85.7 | 3.35 | 69.6 | 85.6 |
| | **Ours** + DualMesh-UDF | **34.0** | **68.6** | **82.8** | 7.81 | **48.2** | **86.5** | **2.6** | **73.3** | **88.4** |

after that. Using a softmax activation, which helps normalizing the otherwise unbound network outputs, the training converges, but the network does not produce significant improvements over iterations, going from an IC of 87.3 to 87.5 after two iterations. Using a sigmoid activation, instead, showed a more steady improvement over iterations, going from 87.6 to 88.5 after one iteration, and 88.9 after an additional one. However, training it with more than one iteration did not show significant improvements. Using a random number of training iterations, instead, helped the network to converge until 5 iterations, achieving similar IC scores but better CD scores ($9.90 \times 10^{-5}$ vs $8.84 \times 10^{-5}$), as shown in the main paper, signifying an overall similar accuracy but better surface retrieval capabilities at high resolutions.

## A.4    Auto-decoder training

For our auto-decoder experiments we used the traditional auto-decoder architecture proposed in DeepSDF [27]. The input meshes are rescaled and centered within a $[-1, 1]^3$ volume, and during the data preparation phase, training points are sampled. For each mesh, 20k points are uniformly sampled within the volume, while 400k points are sampled near the surface. To obtain the surface points, 200k points are first uniformly distributed on the surface, and then small amounts of Gaussian noise are added. Gaussian noise with a mean of 0 and a standard deviation of $\sqrt{0.005}$ is applied to the first 200k surface points, and noise with a mean of 0 and a standard deviation of $\sqrt{0.0005}$ is added to the remaining 200k points. The auto-decoder network consists of 12 layers, each with 1024 hidden nodes and ReLU activations, and latent codes of size 512. It is trained using L1 loss, without regularization or Fourier encoding, for 10k epochs with a batch size of 16. To focus the network's capacity on the surface, the UDF is clamped to 0.1. The Adam optimizer [16] is used with learning rates of 0.0005 for the model and 0.001 for the latent codes, with learning rate decay applied at epochs 1600 and 3500 by a factor of 0.35.

## A.5    Different auto-decoder UDF architecture

To test the robustness of our method to different UDF architectures, we trained an auto-decoder with a different architecture: using a softplus activation function and Eikonal loss with a weight of 0.1, with the rest as in the section above. We show the results on ShapeNet [3] cars in Tab. A.7, along with the highest-scoring baselines from the main experiment. While all methods achieved slightly better performance compared to the ReLU-based architecture, the same conclusions apply. Our method outperforms the tested baselines, particularly so at high resolutions, while also achieving lower standard deviations across the dataset.

Table A.3: **Triangulating auto-decoder-based Neural Unsigned Fields using Marching Cubes-based method.** L2 Mesh Chamfer Distance $\times 10^{-5}$ with 2M sample points (CD), F1 score (F1) and Image Consistency (IC) are reported at varying grid resolutions. The best results are in bold. DCUDF failed to mesh some of the shapes, but its numbers are reported nonetheless. *Resolution is halved for experiments with MGN due to the lower complexity of the shapes.

(a) Mean results.

| Res. | Method | MGN* [1] CD↓ | F1↑ | IC↑ | ShapeNet cars [3] CD↓ | F1↑ | IC↑ | ShapeNet chairs [3] CD↓ | F1↑ | IC↑ | ShapeNet planes [3] CD↓ | F1↑ | IC↑ |
|---|---|---|---|---|---|---|---|---|---|---|---|---|---|
| 128-MC | CAP-UDF [46] | 19.7 | 68.2 | 78.6 | 71.7 | 49.5 | 81.8 | 518 | 51.3 | 68.9 | 19.5 | 68.1 | 79.0 |
| | MeshUDF [13] | 2.79 | 82.0 | 93.3 | 11.9 | 57.9 | 88.7 | 21.8 | 68.0 | 88.7 | 19.4 | 73.5 | 81.0 |
| | DCUDF [14] | 14100 | 3.40 | 3.84 | 8000 | 10.6 | 13.7 | 22700 | 14.4 | 16.8 | 2900 | 28.9 | 24.3 |
| | DCUDF-T [14] | 116 | 3.04 | 86.6 | 66.5 | 55.4 | 86.8 | 2720 | 65.2 | 78.2 | 902 | 69.4 | 78.8 |
| | DCUDF-T-nocut [14] | - | - | - | 14.4 | **60.9** | **89.2** | - | - | - | - | - | - |
| | NSD-UDF + MC [36] | **1.55** | **82.9** | **94.1** | 9.40 | 59.9 | 88.5 | 17.7 | **68.7** | 88.5 | 4.72 | **78.0** | 84.5 |
| | **Ours** + MC | 2.26 | 80.8 | 93.4 | **6.79** | 59.5 | 88.9 | **7.27** | 67.8 | **89.5** | **3.49** | 77.6 | **84.8** |
| 256-MC | CAP-UDF [46] | 3.23 | 85.4 | 91.2 | 48.3 | 58.8 | 86.2 | 223 | 65.5 | 78.5 | 8.95 | 83.2 | 85.0 |
| | MeshUDF [13] | 1.16 | 88.2 | 94.5 | 17.1 | 61.1 | 88.2 | 67.1 | 69.8 | 85.4 | 4.83 | 85.2 | 85.1 |
| | DCUDF [14] | 18200 | 5.17 | 4.12 | 1090 | 50.6 | 75.1 | 7900 | 50.7 | 55.3 | 278 | 82.4 | 80.2 |
| | DCUDF-T [14] | 10.3 | 86.1 | **94.8** | 1080 | 50.3 | 75.0 | 7840 | 50.7 | 55.3 | 797 | 80.1 | 77.9 |
| | DCUDF-T-nocut [14] | - | - | - | 68.4 | 56.4 | 80.7 | - | - | - | - | - | - |
| | NSD-UDF + MC [36] | **0.973** | **88.7** | **94.8** | 14.7 | 61.2 | 87.6 | 53.7 | 69.8 | 85.4 | 3.81 | 86.9 | 85.2 |
| | **Ours** + MC | 1.02 | 87.6 | 94.4 | **7.26** | **63.1** | **89.0** | 10.6 | **70.6** | **89.3** | **2.64** | **88.2** | **86.6** |
| 512-MC | CAP-UDF [46] | 2.14 | 89.1 | 94.1 | 48.6 | 60.2 | 86.4 | 202 | 67.9 | 79.3 | 8.94 | 87.3 | 85.5 |
| | MeshUDF [13] | 1.18 | 89.3 | **94.4** | 136 | 54.4 | 78.8 | 799 | 57.5 | 64.2 | 21.2 | 87.4 | 83.4 |
| | DCUDF [14] | 49.4 | 85.8 | 88.8 | 478 | 52.9 | 80.2 | 7510 | 55.7 | 63.8 | 199 | 84.1 | 81.0 |
| | DCUDF-T [14] | 32.8 | 85.9 | 88.9 | 478 | 52.9 | 80.2 | 7380 | 55.9 | 64.2 | 284 | 83.2 | 79.9 |
| | DCUDF-T-nocut [14] | - | - | - | 61.4 | 58.5 | 84.7 | - | - | - | - | - | - |
| | NSD-UDF + MC [36] | 1.08 | **89.5** | **94.4** | 80.8 | 56.8 | 82.0 | 394 | 62.9 | 71.2 | 13.4 | 88.2 | 83.8 |
| | **Ours** + MC | **0.880** | 89.3 | **94.4** | 12.3 | **63.4** | **88.2** | 36.6 | **71.6** | **86.5** | **3.31** | **90.1** | **86.5** |

(b) Standard deviation for each metric, computed across the dataset.

| Res. | Method | MGN* [1] CD | F1 | IC | ShapeNet cars [3] CD | F1 | IC | ShapeNet chairs [3] CD | F1 | IC | ShapeNet planes [3] CD | F1 | IC |
|---|---|---|---|---|---|---|---|---|---|---|---|---|---|
| 128-MC | CAP-UDF [46] | 12.1 | 7.83 | 3.46 | 51.9 | 12.2 | 5.69 | 649 | 15.4 | 14.7 | 21.3 | 5.63 | 3.45 |
| | MeshUDF [13] | 1.28 | 7.9 | 2.34 | 5.21 | 12.1 | 2.22 | 30.3 | 13.9 | 5.58 | 27.4 | 5.54 | 2.56 |
| | DCUDF [14] | 561 | 0.901 | 1.08 | 5560 | 4.81 | 5.74 | 18800 | 8.78 | 16.0 | 3010 | 7.84 | 7.45 |
| | DCUDF-T [14] | 169 | 1.30 | 3.97 | 57.6 | 11.4 | 4.28 | 5020 | 14.0 | 17.6 | 1400 | 6.72 | 6.79 |
| | DCUDF-T-nocut [14] | - | - | - | 9.59 | 11.1 | 2.49 | - | - | - | - | - | - |
| | NSD-UDF + MC [36] | 0.817 | 7.87 | 1.91 | 5.24 | 12.2 | 2.33 | 28.0 | 13.7 | 5.64 | 2.87 | 6.10 | 2.28 |
| | **Ours** + MC | 0.872 | 7.75 | 2.09 | 3.16 | 11.7 | 1.92 | 8.55 | 13.4 | 4.93 | 1.49 | 6.02 | 1.89 |
| 256-MC | CAP-UDF [46] | 4.83 | 7.23 | 2.37 | 40.8 | 12.6 | 4.46 | 220 | 15.9 | 12.7 | 7.44 | 5.32 | 3.03 |
| | MeshUDF [13] | 0.733 | 6.78 | 1.56 | 11.7 | 11.6 | 2.64 | 74.9 | 14.3 | 8.51 | 4.18 | 5.51 | 2.41 |
| | DCUDF [14] | 12700 | 2.22 | 1.86 | 3250 | 14.0 | 15.2 | 9500 | 17.7 | 19.9 | 664 | 8.45 | 6.52 |
| | DCUDF-T [14] | 12.5 | 7.00 | 1.68 | 3250 | 14.0 | 15.1 | 9440 | 17.6 | 19.9 | 1440 | 8.70 | 7.70 |
| | DCUDF-T-nocut [14] | - | - | - | 51.9 | 12.3 | 7.02 | - | - | - | - | - | - |
| | NSD-UDF + MC [36] | 0.636 | 6.78 | 1.45 | 10.3 | 11.8 | 2.79 | 68.9 | 14.4 | 8.36 | 3.04 | 5.80 | 3.13 |
| | **Ours** + MC | 0.557 | 6.55 | 1.46 | 3.91 | 10.8 | 1.95 | 13.0 | 14.0 | 5.12 | 2.37 | 4.37 | 2.06 |
| 512-MC | CAP-UDF [46] | 3.97 | 6.78 | 1.80 | 40.6 | 12.4 | 4.19 | 217 | 13.8 | 12.2 | 7.65 | 4.03 | 3.20 |
| | MeshUDF [13] | 1.30 | 6.63 | 1.42 | 115 | 12.9 | 8.34 | 964 | 16.3 | 17.8 | 20.0 | 5.30 | 4.67 |
| | DCUDF [14] | 362 | 10.2 | 7.60 | 989 | 14.7 | 12.8 | 9890 | 19.0 | 22.0 | 537 | 8.35 | 4.99 |
| | DCUDF-T [14] | 219 | 9.98 | 7.33 | 989 | 14.7 | 12.8 | 9950 | 19.0 | 22.5 | 774 | 8.53 | 5.93 |
| | DCUDF-T-nocut [14] | - | - | - | 47.4 | 12.4 | 5.51 | - | - | - | - | - | - |
| | NSD-UDF + MC [36] | 1.09 | 6.66 | 1.48 | 65.7 | 12.8 | 6.40 | 442 | 15.1 | 15.8 | 11.3 | 5.08 | 4.38 |
| | **Ours** + MC | 0.563 | 6.49 | 1.27 | 8.41 | 11.2 | 2.48 | 48.5 | 12.7 | 7.55 | 2.83 | 4.08 | 2.73 |

## A.6 Additional figures

We show here additional qualitative results of our method compared to the baselines. In Fig. A.1 we show an additional example at different resolutions compared to the NSD-UDF [36] baseline: the shapes look similar at low resolutions, but at 256 and 512 the baseline cannot retrieve large portions of the surface.

In Fig. A.2 & A.3 we show the same shapes as in Fig. 3 of the main paper, but with different resolutions. In Fig. A.4, A.5 & A.6 we show additional shapes at all tested resolutions. As observed

Table A.4: **Triangulating auto-decoder-based Neural Unsigned Fields using Dual Contouring-based methods.** L2 Mesh Chamfer Distance $\times 10^{-5}$ with 2M sample points (CD), F1 score (F1) and Image Consistency (IC) are reported at varying grid resolutions. The best results are in bold. UNDC failed at resolution 512 due to high GPU memory requirements. *Resolution is halved for experiments with MGN due to the lower complexity of the shapes.

(a) Mean results.

| Res. | Method | MGN* [1] | | | ShapeNet cars [3] | | | ShapeNet chairs [3] | | | ShapeNet planes [3] | | |
|---|---|---|---|---|---|---|---|---|---|---|---|---|---|
| | | CD ↓ | F1 ↑ | IC ↑ | CD ↓ | F1 ↑ | IC ↑ | CD ↓ | F1 ↑ | IC ↑ | CD ↓ | F1 ↑ | IC ↑ |
| 128-DC | UNDC [5] | 1.26 | 86.1 | 93.8 | 17.1 | 60.8 | 86.0 | 91.3 | 66.8 | 78.8 | 3.82 | 81.8 | 85.7 |
| | DualMesh-UDF [41] | 1120 | 61.1 | 60.9 | 1600 | 33.0 | 42.8 | 12400 | 15.0 | 13.0 | 341 | 72.4 | 72.3 |
| | DualMesh-UDF-T [41] | 0.939 | 88.7 | **94.9** | 7.45 | 62.6 | 89.4 | 19.1 | **71.4** | 88.7 | 2.91 | 83.3 | 87.3 |
| | NSD-UDF + DualMesh-UDF [36] | **0.901** | **89.2** | 94.5 | 8.60 | 63.9 | 89.1 | 17.3 | 70.2 | 88.3 | 3.12 | 85.9 | 86.5 |
| | **Ours** + DualMesh-UDF | 0.904 | **89.2** | 94.5 | **5.89** | **64.2** | **89.7** | **7.28** | 70.1 | **89.8** | **2.21** | **86.7** | **88.0** |
| 256-DC | UNDC [5] | 1.32 | 87.9 | 91.2 | 122 | 49.2 | 69.0 | 598 | 50.9 | 55.9 | 21.4 | 81.9 | 78.7 |
| | DualMesh-UDF [41] | 993 | 60.0 | 59.0 | 1440 | 33.3 | 42.6 | 11900 | 15.2 | 13.0 | 158 | 76.1 | 72.5 |
| | DualMesh-UDF-T [41] | 0.899 | **89.7** | **94.7** | 14.3 | 61.8 | 86.6 | 62.2 | 69.0 | 82.4 | 3.26 | 87.8 | 86.8 |
| | NSD-UDF + DualMesh-UDF [36] | 0.838 | 89.5 | 94.3 | 15.6 | 61.8 | 87.0 | 56.0 | 68.3 | 82.4 | 3.72 | 88.3 | 85.4 |
| | **Ours** + DualMesh-UDF | **0.806** | 89.5 | 94.4 | **7.57** | **63.5** | **88.5** | **12.5** | 69.2 | 86.7 | **2.66** | **89.3** | **87.3** |
| 512-DC | UNDC [5] | 6.86 | 83.0 | 81.2 | - | - | - | - | - | - | - | - | - |
| | DualMesh-UDF [41] | 948 | 58.4 | 57.1 | 1560 | 32.2 | 41.1 | 12400 | 14.7 | 12.2 | 172 | 75.6 | 70.6 |
| | DualMesh-UDF-T [41] | 1.25 | **88.8** | **92.7** | 51.7 | 55.4 | 77.9 | 211 | 60.6 | 67.2 | 7.68 | 87.7 | 83.7 |
| | NSD-UDF + DualMesh-UDF [36] | 1.13 | 88.2 | 91.9 | 86.3 | 54.3 | 77.8 | 412 | 58.4 | 63.9 | 13.8 | 86.7 | 82.1 |
| | **Ours** + DualMesh-UDF | **0.899** | 88.2 | 92.3 | **13.5** | **61.0** | **85.1** | **38.9** | **67.1** | 80.1 | **3.47** | **89.0** | **85.2** |

(b) Standard deviation for each metric, computed across the dataset.

| Res. | Method | MGN* [1] | | | ShapeNet cars [3] | | | ShapeNet chairs [3] | | | ShapeNet planes [3] | | |
|---|---|---|---|---|---|---|---|---|---|---|---|---|---|
| | | CD | F1 | IC | CD | F1 | IC | CD | F1 | IC | CD | F1 | IC |
| 128-DC | UNDC [5] | 0.651 | 6.80 | 1.43 | 11.6 | 12.0 | 3.26 | 94.4 | 15.5 | 11.2 | 3.33 | 5.77 | 2.69 |
| | DualMesh-UDF [41] | 2470 | 25.6 | 25.4 | 2090 | 12.0 | 13.2 | 12800 | 12.0 | 11.3 | 995 | 10.8 | 12.2 |
| | DualMesh-UDF-T [41] | 0.561 | 6.51 | 1.36 | 4.17 | 10.5 | 2.10 | 29.3 | 13.5 | 5.94 | 2.58 | 5.34 | 1.94 |
| | NSD-UDF + DualMesh-UDF [36] | 0.546 | 6.48 | 1.50 | 5.14 | 11.4 | 2.21 | 27.2 | 13.3 | 5.55 | 2.76 | 4.96 | 2.33 |
| | **Ours** + DualMesh-UDF | 0.517 | 6.47 | 1.43 | 2.92 | 11.0 | 1.87 | 9.04 | 13.2 | 4.48 | 1.40 | 4.88 | 1.95 |
| 256-DC | UNDC [5] | 1.25 | 6.80 | 2.41 | 97.7 | 11.9 | 9.59 | 716 | 17.6 | 16.1 | 21.1 | 6.89 | 5.63 |
| | DualMesh-UDF [41] | 2280 | 25.9 | 25.5 | 1390 | 11.6 | 12.8 | 12900 | 11.9 | 10.9 | 241 | 9.85 | 11.8 |
| | DualMesh-UDF-T [41] | 0.673 | 6.49 | 1.31 | 9.94 | 10.7 | 3.12 | 72.3 | 13.6 | 8.97 | 2.96 | 3.96 | 2.39 |
| | NSD-UDF + DualMesh-UDF [36] | 0.565 | 6.80 | 1.37 | 11.1 | 11.8 | 3.10 | 71.2 | 13.1 | 8.33 | 3.07 | 4.44 | 3.16 |
| | **Ours** + DualMesh-UDF | 0.516 | 6.81 | 1.26 | 4.21 | 11.1 | 2.22 | 15.9 | 13.0 | 4.98 | 2.57 | 4.08 | 2.30 |
| 512-DC | UNDC [5] | 11.3 | 9.36 | 6.95 | - | - | - | - | - | - | - | - | - |
| | DualMesh-UDF [41] | 2080 | 26.0 | 25.6 | 1960 | 11.4 | 12.5 | 14500 | 11.6 | 10.3 | 267 | 9.88 | 11.7 |
| | DualMesh-UDF-T [41] | 1.36 | 7.14 | 2.51 | 38.5 | 11.5 | 5.96 | 221 | 13.6 | 14.3 | 6.87 | 4.94 | 3.93 |
| | NSD-UDF + DualMesh-UDF [36] | 1.18 | 7.48 | 2.92 | 69.7 | 12.6 | 7.41 | 467 | 14.9 | 15.5 | 11.4 | 5.2 | 4.83 |
| | **Ours** + DualMesh-UDF | 0.616 | 7.48 | 2.54 | 9.32 | 11.4 | 3.03 | 49.6 | 12.3 | 7.63 | 2.83 | 4.36 | 3.10 |

in the main paper, existing methods retrieve most of the surface at lower resolutions, leaving less room for improvement, whereas at higher resolutions our method shows a significant advantage.
In Fig. A.7, we show the meshing on the CAP-L 3D scene [46] used in Fig. 4 of the main text in greater size to better appreciate the details, *e.g.,* the base of the statues.

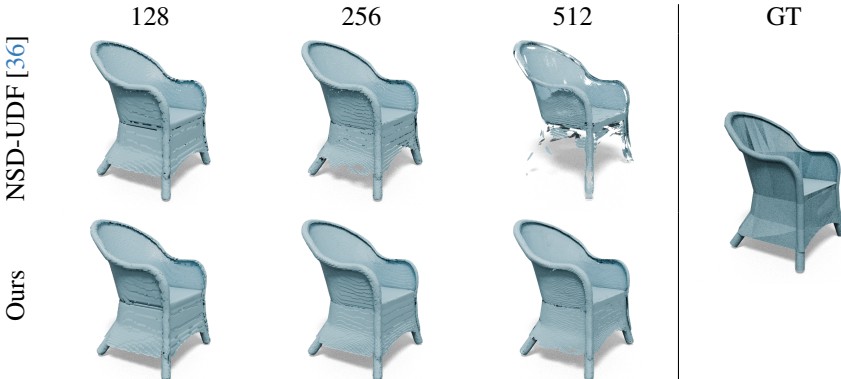

Figure A.1: **Meshing at different resolutions, additional examples.** While NSD-UDF [36] retrieves most of the surface well at a low resolution, it struggles at higher ones. In contrast, our method, recovers the surface well at all resolutions. We use Marching Cubes with both methods.

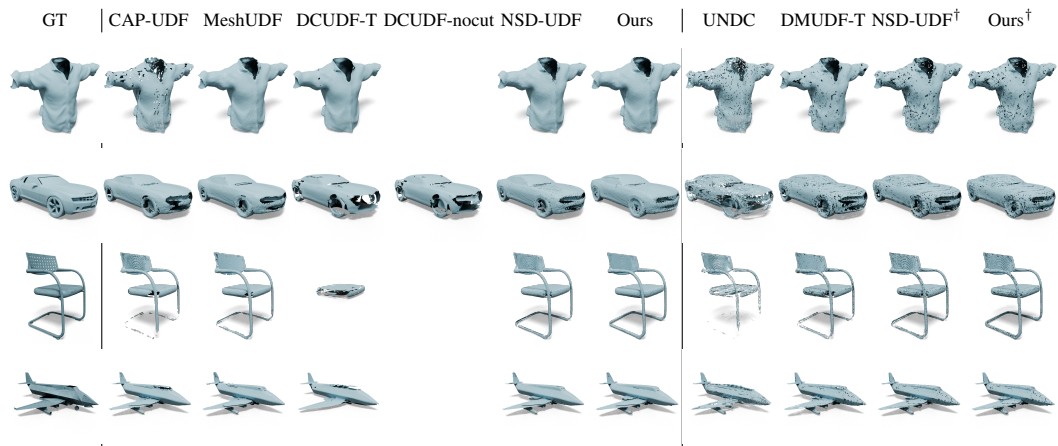

Figure A.2: **Qualitative comparison at resolution 256.** Surface meshing results of auto-decoder-based neural UDFs with all methods at resolution of 256 (and 128 for MGN). $^\dagger$ indicates that the method is combined with DMUDF. The shapes are the same as in the main text.

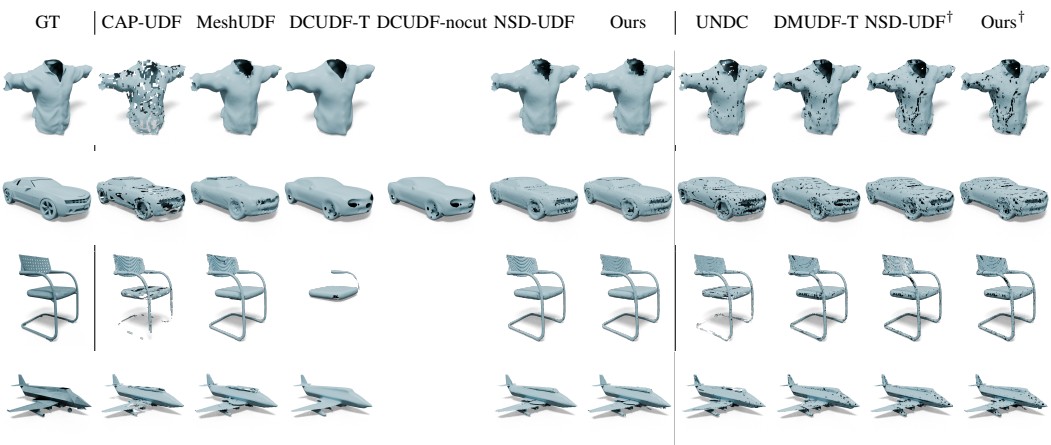

Figure A.3: **Qualitative comparison at resolution 128.** Surface meshing results of auto-decoder-based neural UDFs with all methods at resolution of 128 (and 64 for MGN). $^\dagger$ indicates that the method is combined with DMUDF. The shapes are the same as in the main text.

Table A.5: **Neural Unsigned Distance Fields from point clouds (MC-based methods).** L2 Mesh Chamfer Distance $\times 10^{-5}$ with 2M sample points (CD), F1 score (F1) and Image Consistency (IC) are reported at varying grid resolutions. The best results are in bold.

(a) Mean results.

| Res. | Method | CAP-L scenes [46] | | | CAP-L cars [46] | | | DiffUDF cars [11] | | |
|---|---|---|---|---|---|---|---|---|---|---|
| | | CD ↓ | F1 ↑ | IC ↑ | CD ↓ | F1 ↑ | IC ↑ | CD ↓ | F1 ↑ | IC ↑ |
| 128-MC | CAP-UDF [46] | 4.27 | 68.0 | 83.8 | 11.4 | 48.5 | 85.3 | 7.96 | 60.1 | 86.2 |
| | MeshUDF [13] | 4.40 | 68.2 | 84.5 | 11.9 | 49.7 | 86.1 | 9.52 | 61.4 | 87.0 |
| | DCUDF-T [14] | 279 | 59.7 | 84.0 | 131 | 45.0 | 83.7 | 99.4 | 53.2 | 87.0 |
| | DCUDF-T-nocut [14] | - | - | - | 10.7 | 51.3 | **87.6** | 15.3 | 58.6 | **90.2** |
| | NSD-UDF + MC [36] | **3.34** | **69.8** | **86.7** | 11.1 | 51.7 | 86.3 | **4.86** | **66.0** | 87.8 |
| | **Ours** + MC | 3.44 | 69.3 | 85.3 | **9.76** | **51.8** | 86.8 | 6.65 | 63.6 | 87.5 |
| 256-MC | CAP-UDF [46] | 3.65 | 70.1 | 86.1 | 10.1 | 53.4 | 86.4 | 4.04 | 69.0 | 86.6 |
| | MeshUDF [13] | 3.42 | 69.6 | 86.3 | 10.2 | 52.3 | 86.1 | 5.61 | 66.8 | 87.5 |
| | DCUDF-T [14] | 4.75 | 69.9 | 84.6 | 42.1 | 50.7 | 84.7 | 106 | 63.8 | 84.6 |
| | DCUDF-T-nocut [14] | - | - | - | 12.0 | 53.1 | 85.7 | 6.37 | 68.8 | 87.1 |
| | NSD-UDF + MC [36] | 3.30 | 70.3 | 86.3 | 11.5 | 52.9 | 86.1 | **3.85** | 70.6 | 86.7 |
| | **Ours** + MC | **3.07** | **71.4** | **86.6** | **9.93** | **54.0** | **86.8** | 4.23 | 69.2 | **88.0** |
| 512-MC | CAP-UDF [46] | 3.57 | 70.5 | 86.1 | 10.8 | 53.8 | 86.2 | 4.87 | 69.3 | 84.0 |
| | MeshUDF [13] | 4.49 | 69.9 | 84.5 | 10.9 | 52.7 | 85.7 | 4.25 | 68.8 | 86.2 |
| | DCUDF-T [14] | 140 | 68.9 | 83.3 | 37.0 | 50.9 | 85.0 | 727 | 61.8 | 78.3 |
| | DCUDF-T-nocut [14] | - | - | - | 13.5 | 51.5 | 84.4 | 6.64 | 68.2 | 86.4 |
| | NSD-UDF + MC [36] | 3.72 | 70.0 | 84.1 | 12.3 | 52.5 | 85.6 | 5.92 | 67.1 | 81.1 |
| | **Ours** + MC | **3.08** | **71.8** | **86.8** | **10.5** | **54.1** | **86.4** | **3.48** | **71.9** | **87.8** |

(b) Standard deviation for each metric, computed across the dataset.

| Res. | Method | CAP-L scenes [46] | | | CAP-L cars [46] | | | DiffUDF cars [11] | | |
|---|---|---|---|---|---|---|---|---|---|---|
| | | CD | F1 | IC | CD | F1 | IC | CD | F1 | IC |
| 128-MC | CAP-UDF [46] | 1.98 | 19.9 | 2.43 | 4.71 | 12.2 | 3.07 | 3.88 | 11.2 | 2.53 |
| | MeshUDF [13] | 1.38 | 20.1 | 1.42 | 5.77 | 13.9 | 3.13 | 5.23 | 13.4 | 2.97 |
| | DCUDF-T [14] | 385 | 20.1 | 6.53 | 318 | 12.9 | 5.91 | 109 | 12.0 | 5.37 |
| | DCUDF-T-nocut [14] | - | - | - | 5.37 | 12.6 | 3.06 | 12.5 | 12.5 | 1.65 |
| | NSD-UDF + MC [36] | 1.67 | 20.8 | 2.20 | 6.91 | 14.2 | 3.31 | 3.54 | 12.6 | 2.69 |
| | **Ours** + MC | 1.67 | 20.0 | 1.8 | 6.26 | 13.3 | 2.75 | 5.29 | 12.1 | 2.30 |
| 256-MC | CAP-UDF [46] | 1.91 | 21.1 | 1.94 | 6.65 | 14.4 | 3.46 | 2.35 | 12.6 | 3.41 |
| | MeshUDF [13] | 1.79 | 21.3 | 2.03 | 5.65 | 14.9 | 3.48 | 3.06 | 13.3 | 2.78 |
| | DCUDF-T [14] | 3.71 | 21.8 | 2.99 | 46.7 | 15.5 | 4.38 | 202 | 14.3 | 7.80 |
| | DCUDF-T-nocut [14] | - | - | - | 6.85 | 15.2 | 4.13 | 3.66 | 13.4 | 3.40 |
| | NSD-UDF + MC [36] | 1.76 | 21.5 | 2.63 | 7.27 | 15.1 | 3.69 | 2.47 | 13.0 | 3.56 |
| | **Ours** + MC | 1.60 | 20.3 | 2.04 | 6.63 | 14.4 | 3.11 | 3.31 | 11.9 | 2.24 |
| 512-MC | CAP-UDF [46] | 1.98 | 21.3 | 2.44 | 7.26 | 15 | 3.64 | 3.14 | 14.2 | 5.29 |
| | MeshUDF [13] | 3.38 | 21.5 | 3.45 | 6.35 | 15.3 | 3.94 | 2.46 | 13.3 | 3.30 |
| | DCUDF-T [14] | 191 | 21.3 | 4.07 | 28.2 | 15.3 | 4.21 | 1310 | 15.8 | 12.2 |
| | DCUDF-T-nocut [14] | - | - | - | 7.45 | 15.3 | 5.10 | 3.78 | 13.5 | 3.57 |
| | NSD-UDF + MC [36] | 2.07 | 21.1 | 3.21 | 7.66 | 15.4 | 4.07 | 3.95 | 15.6 | 7.05 |
| | **Ours** + MC | 1.63 | 20.4 | 2.54 | 6.99 | 15.0 | 3.48 | 2.15 | 12.4 | 2.73 |

Table A.6: **Neural Unsigned Distance Fields from point clouds (DC-based methods).** L2 Mesh Chamfer Distance $\times 10^{-5}$ with 2M sample points (CD), F1 score (F1) and Image Consistency (IC) are reported at varying grid resolutions. The best results are in bold. UNDC failed at resolution 512 due to its large GPU memory requirements.

(a) Mean results.

| Res. | Method | CAP-L scenes [46] | | | CAP-L cars [46] | | | DiffUDF cars [11] | | |
|---|---|---|---|---|---|---|---|---|---|---|
| | | CD ↓ | F1 ↑ | IC ↑ | CD ↓ | F1 ↑ | IC ↑ | CD ↓ | F1 ↑ | IC ↑ |
| 128-DC | UNDC [5] | **3.34** | 69.7 | 84.4 | 12.7 | 52.3 | 84.8 | 4.76 | 60.9 | 87.6 |
| | DualMesh-UDF [41] | 53.5 | 63.4 | 69.5 | 11.3 | 54.0 | 85.0 | 5.77 | 62.5 | 84.3 |
| | NSD-UDF + DualMesh-UDF [36] | 33.6 | 69.8 | **84.5** | 7.01 | 56.0 | 87.0 | **3.51** | **68.3** | **88.3** |
| | **Ours** + DualMesh-UDF | 33.6 | **70.0** | 84.4 | **6.73** | 56.2 | **87.2** | 4.37 | 66.7 | 88.2 |
| 256-DC | UNDC [5] | **7.85** | **69.1** | 76.9 | 9.89 | 52.9 | 83.9 | 17.5 | 58.3 | 81.2 |
| | DualMesh-UDF [41] | 50.4 | 63.0 | 68.3 | 10.8 | 53.2 | 84.3 | 5.43 | 61.1 | 82.5 |
| | NSD-UDF + DualMesh-UDF [36] | 34.3 | 68.9 | **83.4** | 8.42 | 54.0 | 86.2 | **3.08** | **71.5** | 87.7 |
| | **Ours** + DualMesh-UDF | 220 | 49.9 | 79.6 | **7.95** | **54.6** | **86.7** | 3.36 | 70.4 | **88.3** |
| 512-DC | UNDC [5] | - | - | - | - | - | - | - | - | - |
| | DualMesh-UDF [41] | 51.7 | 59.8 | 61.8 | 11.4 | 50.6 | 83.1 | 5.69 | 58.2 | 79.0 |
| | NSD-UDF + DualMesh-UDF [36] | 35.2 | 66.5 | 77.9 | 10.1 | 51.7 | 85.4 | 3.99 | 68.8 | 85.1 |
| | **Ours** + DualMesh-UDF | **34.0** | **68.6** | **82.8** | **9.15** | **52.5** | **86.2** | **3.05** | **72.3** | **87.9** |

(b) Standard deviation for each metric, computed across the dataset.

| Res. | Method | CAP-L scenes [46] | | | CAP-L cars [46] | | | DiffUDF cars [11] | | |
|---|---|---|---|---|---|---|---|---|---|---|
| | | CD | F1 | IC | CD | F1 | IC | CD | F1 | IC |
| 128-DC | UNDC [5] | 2.55 | 20.2 | 3.97 | 13.4 | 13.0 | 3.27 | 2.22 | 13.1 | 2.14 |
| | DualMesh-UDF [41] | 35.7 | 18.9 | 7.07 | 6.58 | 13.8 | 4.27 | 2.77 | 13.7 | 3.42 |
| | NSD-UDF + DualMesh-UDF [36] | 42.8 | 17.6 | 0.739 | 3.98 | 13.1 | 2.88 | 2.04 | 11.7 | 2.14 |
| | **Ours** + DualMesh-UDF | 43.0 | 17.3 | 0.803 | 3.70 | 12.8 | 2.60 | 3.02 | 11.8 | 2.00 |
| 256-DC | UNDC [5] | 8.26 | 21.8 | 9.61 | 5.38 | 14.4 | 3.88 | 30.5 | 10.3 | 1.65 |
| | DualMesh-UDF [41] | 35.7 | 19.5 | 7.01 | 6.34 | 14.7 | 4.60 | 2.66 | 13.9 | 3.16 |
| | NSD-UDF + DualMesh-UDF [36] | 43.3 | 18.6 | 0.403 | 4.82 | 14.3 | 3.51 | 1.96 | 12.5 | 2.82 |
| | **Ours** + DualMesh-UDF | 243 | 28.1 | 6.58 | 4.43 | 14.0 | 3.11 | 2.34 | 12.0 | 2.17 |
| 512-DC | UNDC [5] | - | - | - | - | - | - | - | - | - |
| | DualMesh-UDF [41] | 35.0 | 19.9 | 8.32 | 6.45 | 14.6 | 4.86 | 2.68 | 13.5 | 3.03 |
| | NSD-UDF + DualMesh-UDF [36] | 42.3 | 18.6 | 2.04 | 5.74 | 14.7 | 4.04 | 2.50 | 14.0 | 3.80 |
| | **Ours** + DualMesh-UDF | 43.2 | 18.2 | 0.0709 | 5.05 | 14.6 | 3.53 | 1.89 | 12.4 | 2.86 |

Table A.7: **Triangulating a Softplus-based auto-decoder.** L2 Chamfer Distance $\times 10^{-5}$ with 2M sample points (CD), F1 score (F1) and Image Consistency (IC) are reported at varying grid resolutions on the ShapeNet [3] cars dataset. The best results are in bold.

| Res. | Method | Median | | | Mean | | | Std | | |
|---|---|---|---|---|---|---|---|---|---|---|
| | | CD ↓ | F1 ↑ | IC ↑ | CD ↓ | F1 ↑ | IC ↑ | CD | F1 | IC |
| 128-MC | CAP-UDF [46] | 60.9 | 52.5 | 80.7 | 69.2 | 49.9 | 80.1 | 56.6 | 10.5 | 4.51 |
| | MeshUDF [13] | 8.06 | 61.4 | 88.2 | 10.0 | 61.3 | 88.6 | 6.40 | 11.6 | 2.17 |
| | NSD-UDF + MC [36] | 5.95 | **64.9** | 88.6 | 7.69 | **63.5** | 88.7 | 5.13 | 11.9 | 2.26 |
| | **Ours** + MC | **4.80** | 64.0 | **89.0** | **5.85** | 63.1 | **89.1** | 3.53 | 11.3 | 1.94 |
| 256-MC | CAP-UDF [46] | 23.7 | 65.2 | 86.8 | 30.0 | 63.9 | 86.0 | 23.9 | 12.3 | 3.96 |
| | MeshUDF [13] | 13.4 | 67.9 | 88.4 | 15.1 | 66.3 | 88.3 | 10.4 | 11.7 | 2.56 |
| | NSD-UDF + MC [36] | 8.63 | 69.3 | 88.3 | 10.4 | 66.7 | 88.0 | 8.94 | 11.9 | 2.57 |
| | **Ours** + MC | **4.95** | **70.4** | **89.6** | **6.28** | **68.2** | **89.3** | 4.69 | 11.3 | 1.89 |
| 512-MC | CAP-UDF [46] | 23.8 | 67.3 | 87.2 | 30.3 | 65.8 | 86.4 | 24.6 | 12.6 | 3.97 |
| | MeshUDF [13] | 58.3 | 62.3 | 77.9 | 69.5 | 59.6 | 77.4 | 45.2 | 13.2 | 7.15 |
| | NSD-UDF + MC [36] | 21.2 | 68.0 | 85.8 | 24.6 | 65.0 | 84.8 | 19.9 | 12.8 | 4.44 |
| | **Ours** + MC | **8.83** | **71.4** | **88.9** | **10.7** | **68.7** | 88.7 | 9.25 | 11.6 | 2.37 |

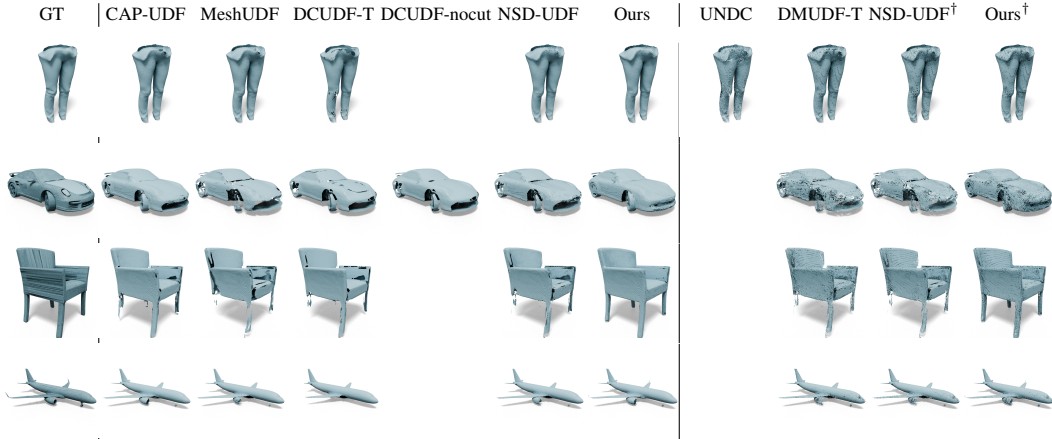

Figure A.4: **Additional qualitative comparison at resolution 512.** Surface meshing results of auto-decoder-based neural UDFs with all methods at resolution of 512 (and 256 for MGN). UNDC failed at resolution 512 due to high GPU memory requirements. † indicates that the method is combined with DMUDF.

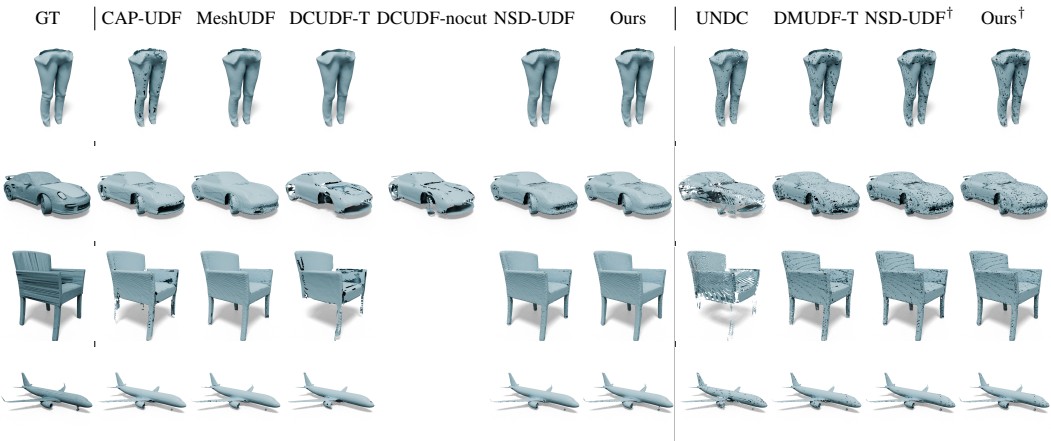

Figure A.5: **Additional qualitative comparison at resolution 256.** Surface meshing results of auto-decoder-based neural UDFs with all methods at resolution of 256 (and 128 for MGN). † indicates that the method is combined with DMUDF.

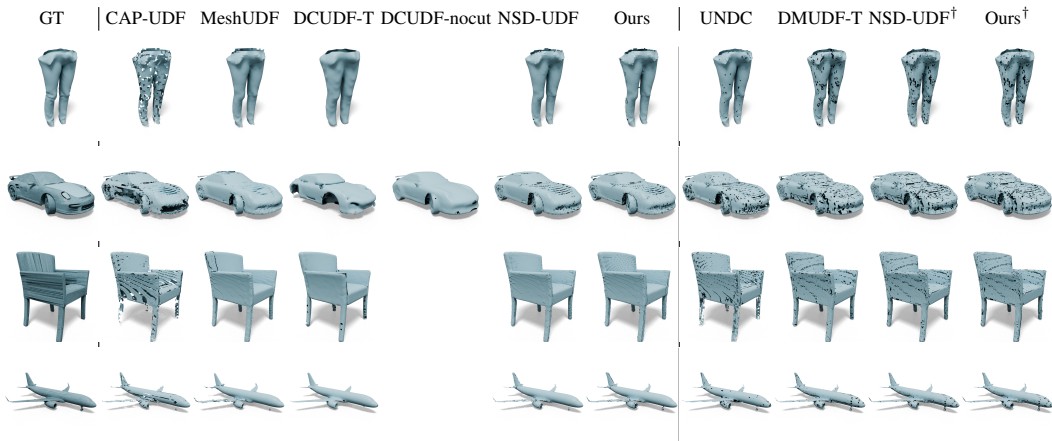

Figure A.6: **Additional qualitative comparison at resolution 128.** Surface meshing results of auto-decoder-based neural UDFs with all methods at resolution of 128 (and 64 for MGN). † indicates that the method is combined with DMUDF.

| GT | CAP-UDF | MeshUDF | DCUDF-T | NSD-UDF |
| Ours | DMUDF† | NSD-UDF† | Ours† |

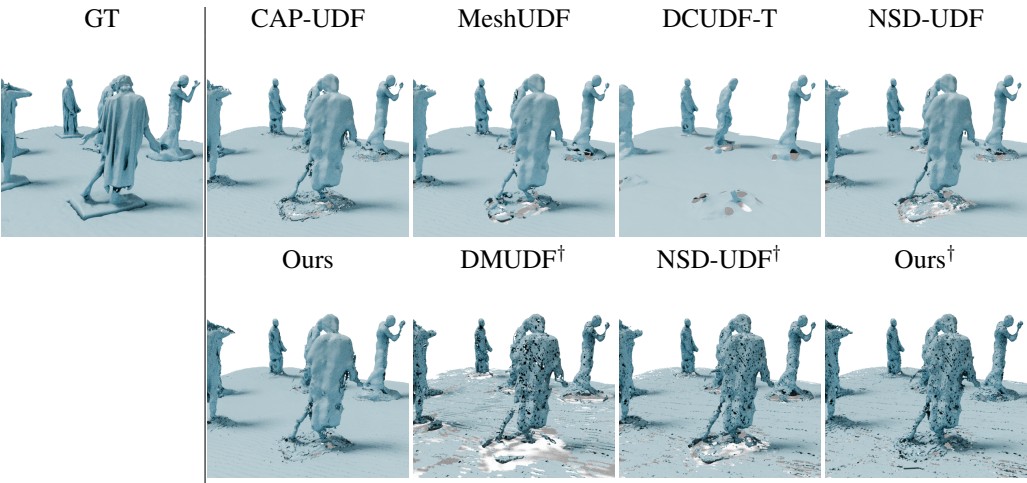

Figure A.7: **Meshing of a 3D scene from CAP-L [46].** Surface meshing results of the "Burghers" scene with all methods at resolution of 512. UNDC failed at resolution 512 due to high GPU memory requirements. † indicates that the method is Dual Contouring-based, otherwise it is Marching Cubes-based.

## A.7  DoubleCoverUDF [14] tuning and min-cut

To achieve better results with DCUDF [14] compared to its default parameters, we have run the algorithm 3 times per experiment and per resolution, with different parameters, and we have selected the best run in each scenario. Additionally, in Fig. A.8, we present the meshing results of DCUDF [14] on UDFs from different models on cars, both with and without the final min-cut step, at resolutions of 128 and 512. Without the min-cut, more surfaces are retained, though they are double-layered. Moreover, the overall meshing quality deteriorates at higher resolutions, following the trend observed for the other baselines.

## A.8  Meshing ground-truth UDF

Although our method is designed to handle imperfect UDFs, we also evaluate it on true UDFs computed directly from ground-truth meshes to verify that it does not introduce unwanted artifacts. We report results on the ShapeNet [3] cars dataset in Tab. A.8 (top), and visualize several triangulations at a resolution of 512 in Fig. A.9. As observed, our method does not introduce significant artifacts; however, as expected, multiple iterations provide no benefit in this setting.
In the same table, we also compute the number of holes as surface boundaries, similarly to the description provided by DCUDF [14], at resolution 512. We report the average. We notice that the original meshes have a large number of boundaries because they contain multiple detailed components and inner structures. None of the methods faithfully recovers the correct mesh topology. Methods that rely directly on MC triangulation, such as CAP-UDF [46], NSD-UDF+MC [36] and Ours+MC, tend to suffer from micro-holes and gaps between some of the faces. MeshUDF uses a heuristic specifically designed to reduce this behavior and connect as many portions of the surface as possible, explaining the lower number of holes. DCUDF-T [14] starts from an inflated mesh, so it generally tends to have fewer holes. Simple postprocessing steps can be applied to all methods to improve the final mesh quality. We take as an example the first of the ShapeNet Cars (object 100715345ee54d7ae38b52b4ee9d36a3), and we apply Trimesh-based postprocessing (fill small holes, merge close vertices, remove spurious faces), showing that all methods improve.

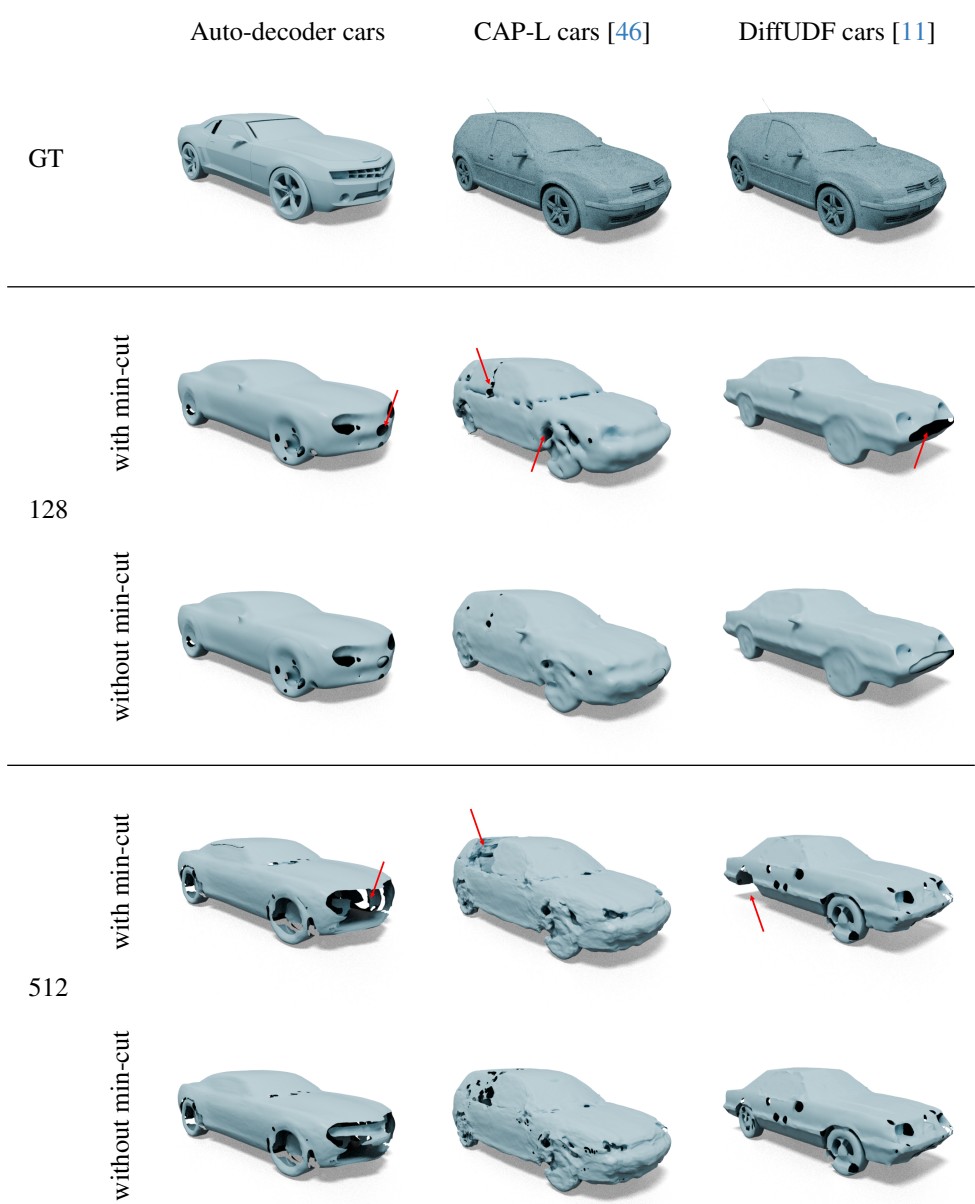

Figure A.8: **DoubleCoverUDF min-cut.** Comparing reconstructions of DCUDF [14] with and without the min-cut step at resolution 128 (middle) and 512 (bottom).

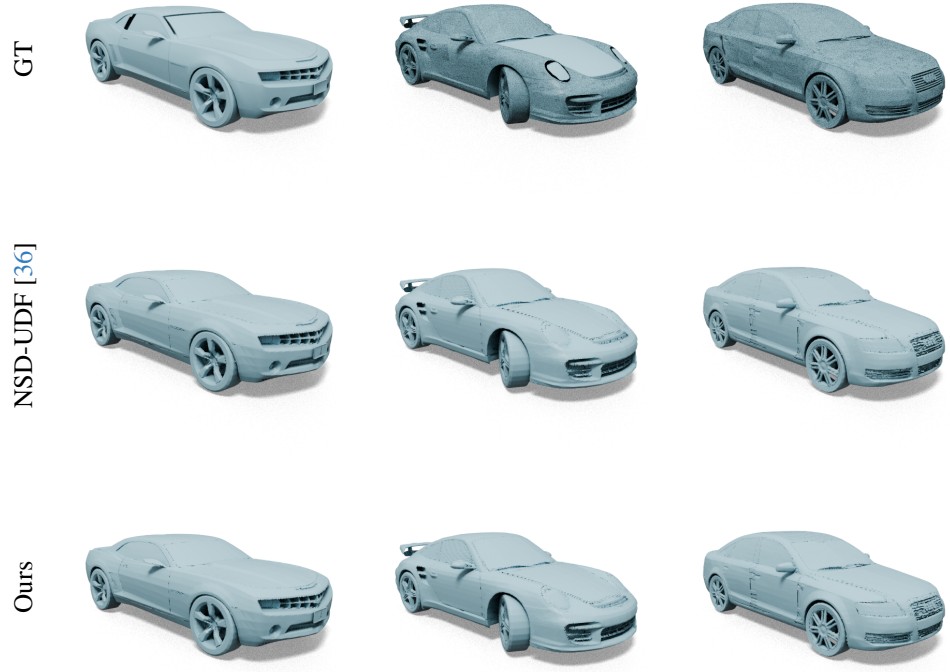

Figure A.9: **Triangulating ground-truth UDF.** Surface meshing results on the ground-truth UDF in comparison to NSD-UDF [36]. Our method does not introduce significant unwanted artifacts.

Table A.8: **Triangulating ground-truth UDFs.** Results are reported on the ShapeNet [3] cars dataset, using the true UDF computed from the mesh. We compare against NSD-UDF + MC [36] at multiple resolution with the Median L2 Chamfer Distance $\times 10^{-5}$ with 2M sample points (Chamfer-Distance), and against all MC-based methods with the average number of holes, as surface boundaries at resolution 512(Number of holes). For the latter, we also report results on the first car of the dataset, before and after post-processing the mesh.

(a) Chamfer-Distance

| Method | Res. 128 | Res. 256 | Res. 512 |
|---|---|---|---|
| NSD-UDF + MC [36] | 1.34 | 0.230 | 0.0300 |
| Ours + MC at iter. 1 | 1.36 | 0.244 | 0.0294 |
| Ours + MC at iter. 6 | 1.40 | 0.231 | 0.0300 |

(b) Number of holes at resolution 512

| | ShapeNet cars [3] | Car 1 (before post-process.) | Car 1 (post-processed) |
|---|---|---|---|
| GT | 783 | 1552 | - |
| CAP-UDF [46] | 10024 | 13265 | 2125 |
| MeshUDF [13] | 428 | 997 | 240 |
| DCUDF-T [14] | 7 | 6 | 6 |
| DCUDF-T-nocut [14] | 2.65 | 2 | 2 |
| NSD-UDF+MC [36] | 8280 | 8885 | 1387 |
| Ours+MC | 9284 | 9808 | 1604 |

