# OpenReview forum: "High Resolution UDF Meshing via Iterative Networks"
_NeurIPS.cc/2025/Conference — NeurIPS 2025 poster_

### Official Review · Reviewer_YyyE · 2025-06-07

**Clarity:** 2
**Significance:** 1
**Originality:** 2
**Rating:** 4
**Confidence:** 4

**Summary:**

The authors present a framework that reconstruct the mesh from a learned unsigned distance field (UDF) network. Given a learned UDF network, this paper iteratively query the UDF network on a grid and enforces sign configuration consistency with its neighboring cells. Doing so helps close the holes due to UDF networks usually cannot reach true zero in their UDF predictions.

**Questions:**

I cannot accept the paper with an inherent unfair comparison. If the authors could demonstrate the UDF extraction results with more advanced UDF learning methods, and show that they exhibit better qualities over other UDF mesh extraction methods, I would reconsider my rating.

**Ethical Concerns:**

["NO or VERY MINOR ethics concerns only"]

**Final Justification:**

After multiple rounds of discussion and opinions from other reviewers I decided to raise my score to resolve the discrepancy between me and other reviewers. I am willing to accept the paper **conditionally** if the authors could:
1. Demonstrate qualitative visualisations on GT UDF.
2. Complete the experiments with DiffUDF and CAP-UDF backbones, and incorporate the qualitative & quantitative results to the main text.
3. Demonstrate the results of DCUDF **without mini-cut**, and incorporate the qualitative visualisations to the main text.

I would also encourage that the authors conduct experiments on watertight models for topology evaluations. The ground truth meshes from ShapeNet Cars is known to contain many fake holes. Therefore I suggest using watertight meshes for that.

I thank the authors' passionate discussion and appreciate their efforts to defend their work.

**Limitations:**

Yes.

**Paper Formatting Concerns:**

No major formatting concerns.

**Quality:**

1

**Strengths And Weaknesses:**

**Strengths**
1. The authors present an improvement over various previous works. There is a catch though, see below.

**Weaknesses**
1. The authors use an autoencoder to learn the UDF field, and such choice is subject to serious question. For example in the teaser image (Figure 1), the authors claim that neural UDFs often fail to reach zero. While this is often the case, the quality of such learned UDF field should not be bad enough to fail to reconstruct the nosewheel of the aircraft. This brings the question of the intrinsic quality of the learned UDF in this paper. The authors follow the architecture of DeepSDF dating back to 2019. In recent years there have been multiple UDF learning methods -- CAP-UDF [1] (the learning part), GeoUDF [2], LevelsetUDF [3], DUDF [4], DEUDF [5], just to name a few. It is unclear whether the bad results (from CAP-UDF, DCUDF, etc) shown in this paper are the fault of the mesh extraction methods or the learned UDF field, and if the latter is the case, the comparison of the paper is unfair. While in supplementary material the authors present another UDF network setup (Section A.5), it is still an autoencoder.
2. I am also very concerned about the authors' misrepresentation of other works. Specifically, the authors claim that GaussianUDF [6] also follows the autoencoder framework used in this paper to represent the UDF (L188; suppl. L46-47), which seems completely untrue -- their network follows NeuralUDF [7], which is a simple MLP with positional encoding. See their paper Section 4.1 "Details" "The UDF $f$ ... 8-layer MLP with 256 hidden units ... apply positional encoding ..."

**Others**
1. The paper heavily uses the term "nerual UDF", but since there is a well-known work literally named "NeuralUDF" [7], it only adds to the confusion.

[1] Zhou et al. "CAP-UDF: Learning Unsigned Distance Functions Progressively from Raw Point Clouds with Consistency-Aware Field Optimization" (NeurIPS 2022 / TPAMI)

[2] Ren et al. "GeoUDF: Surface Reconstruction from 3D Point Clouds via Geometry-guided Distance Representation" (ICCV 2023)

[3] Zhou et al. "Learning a More Continuous Zero Level Set in Unsigned Distance Fields through Level Set Projection" (ICCV 2023)

[4] Fainstein et al. "DUDF: Differentiable Unsigned Distance Fields with Hyperbolic Scaling" (CVPR 2024)

[5] Xu et al. "Details Enhancement in Unsigned Distance Field Learning for High-fidelity 3D Surface Reconstruction" (AAAI 2025)

[6] Li et al. "GaussianUDF: Inferring Unsigned Distance Functions through 3D Gaussian Splatting" (To appear in CVPR 2025)

[7] Long et al. "NeuralUDF: Learning Unsigned Distance Fields for Multi-view Reconstruction of Surfaces with Arbitrary Topologies" (CVPR 2023)

---

> ### Author Rebuttal · Authors · 2025-07-31
>
> Before addressing the reviewer's specific concerns, we would like to clear a potential misunderstanding. The review refers several times to our experiments as using an "autoencoder". This may simply be a typo or shorthand, but just to clarify: we actually use an auto-decoder framework, not an autoencoder. Specifically, our network learns a shape distribution from multiple training shapes using per-shape latent codes and a shared MLP - in the spirit of DeepSDF and subsequent works [4,5]. Thus, there is no encoder involved in our setup. This is a valid choice because auto-decoders remain a widely used and well-established approach [4,5,6,7] in shape modeling, particularly when the focus is on high-fidelity reconstruction and controllable latent space exploration. This has proven valuable in many applications such as interpolation between shapes, shape completion, generative modeling, or shape optimization - all of which commonly rely on mesh extraction, which is the focus of our work. We mention this only to avoid confusion, as autoencoders and auto-decoders differ significantly in architecture and training objectives. We appreciate the reviewer's detailed engagement with our method and hope this clears up some ambiguity.
>
> ***
>
> > **Q1**: The authors use an autoencoder to learn the UDF field, and such choice is subject to serious questions. Why does the teaser (Figure 1) show such a poor reconstruction (e.g., missing nosewheel), given that neural UDFs, while often failing to reach zero, usually perform better?
>
> **A1**: We thank the reviewer for the question. The missing surface in Figure 1 showcases the more significant failure modes that we encountered in our experiments. Other less pronounced examples are visible in Figures 3-4-5, where smaller portions of the shapes are missing. More precise learning-based UDFs can indeed yield better shapes overall, but reaching a precise zero remains difficult, as also noted by the reviewer. So, we have shown a fairly extreme case for exposition purposes but similar situations are sure to arise, maybe to a less extreme extent or at higher resolutions. This is shown by Section A.5 of the supplementary material, which tests our pipeline on a more precise UDF. We also report additional results on an even more precise single-shape UDFs at the end of this rebuttal.
>
> Our intention is to highlight the challenge we deal with: at higher resolutions the UDF must be extremely precise near zero to localize the surface accurately, while the gradients become increasingly noisy near the surface. These factors together make high-resolution meshing particularly difficult  -  a key motivation for our iterative refinement scheme. We will clarify this more clearly in the revision and provide results from both auto-decoder and recent single-shape approaches for completeness.
>
> ***
>
> > **Q2**: Does the paper rely on naive UDF learning methods (e.g., DeepSDF), and could more modern approaches (e.g., CAP-UDF, GeoUDF, DUDF) lead to better results?
>
> **A2**: Our choice to use a DeepSDF-style auto-decoder is motivated by its use both in prior works on meshing methods (like NSD-UDF, MeshUDF) and in prior works on surface representation [4,5,6,7], allowing for controlled and direct comparison of mesh extraction methods on a widespread scenario. That said, we agree that recent UDF learning methods (e.g., CAP-UDF, DUDF, GeoUDF) can offer improved field quality and smoother zero-level sets in certain applications. To address this, at the end of this rebuttal and in the rebuttal for reviewer EmVW, we provide the results on 2 ShapeNet cars and 3 complex scenes from 3D Scene using the CAP-UDF learning pipeline and the point clouds provided by its authors. We find that the results of Copyroom and Lounge are not correctly normalized, resulting in terrible metrics for all of the meshing methods we tested, so we remove them from the 5 originally provided 3D Scene scenes, to avoid skewing the mean metrics. We report results for all MC-based methods tested in the manuscript.
>
> ***
>
> > **Q3**: If the bad results are because the UDF field is noisy, the comparison of the paper is unfair.
>
> **A3**: We respectfully but strongly disagree with this. Our method is a mesh extraction technique, and is agnostic to how the UDF is learned. All methods in our comparisons are evaluated using the exact same input UDF fields, ensuring a controlled and fair comparison focused purely on extraction performance.
>
> Given that learning a perfect UDF is still widely regarded as an open challenge, it is crucial for a meshing method to perform robustly under realistic, imperfect conditions. In that light, the fact that our method consistently outperforms others even with noisy UDF inputs further demonstrates its effectiveness and practical utility.
>
> However we agree that the paper could be strengthened by additional results from single-shape UDF networks such as CAP-UDF, which we include at the end of this rebuttal and in the rebuttal for reviewer EmVW, as discussed above.
>
> ***
>
> > **Q4**: The authors claim that GaussianUDF [6] also follows the autoencoder framework used in this paper to represent the UDF (L188; suppl. L46-47), which seems completely untrue.
>
> **A4**: The sentences in our manuscript ("we train an autodecoder as in [DeepSDF, NSD-UDF, GaussianUDF]" and "we used the traditional autodecoder architecture proposed in DeepSDF and used also in [NSD-UDF, GaussianUDF]") are wrong. We thank the reviewer for pointing it out and we will rewrite them to reflect what we meant. GaussianUDF does not use an auto-decoder, but it uses a ReLU-based MLP, which is the same network architecture that we also use. What we meant is that we train an auto-decoder as in DeepSDF and NSD-UDF, using a similar network architecture also used in recent UDF methods such as GaussianUDF to emphasize that, while introduced a few years ago, ReLU-based MLPs are still a popular choice for UDFs.
>
> ***
>
> > **Q5**: The paper heavily uses the term "nerual UDF", but since there is a well-known work literally named "NeuralUDF", it only adds to the confusion.
>
> **A5**: Thank you for pointing this out. We used this term because it has been used in existing literature (DualMesh-UDF and NSD-UDF). To avoid ambiguity, we will revise the terminology in the paper to use clearer alternatives like "learned UDF" or "MLP-based UDF", and explicitly distinguish our usage from the NeuralUDF work.
>
> ***
>
> ## Surface extraction from CAP-UDF[1,3]
>
> *Note: CAP-UDF [1,3] defines a UDF architecture and training as well as a meshing algorithm. To avoid confusion, we refer here to the learning part as CAP-UDF-L, and to the meshing part as CAP-UDF-M, which has been referred to as CAP-UDF in the manuscript.*
>
> We train CAP-UDF-L [1] networks following the official code repository on the point clouds provided by the authors: 2 cars from ShapeNet, which are not part of our other experiments, using 10k points as described in [1]; and 3 scenes from 3D Scene [2], using the 100 $points/m^2$ scenario described in [1]. We compare Marching Cubes-based methods using the same metrics as the main paper. Quantitative results are shown at the end of the rebuttal for reviewer EmVW.
>
> On cars, which are learned using a relatively dense input point cloud, most methods perform quite well at resolution 128 and 256, with NSD-UDF and our own pipeline closely competing. At 512 a similar scenario as in the main paper arises, with the competing methods sometimes losing accuracy compared to their scores at 256. We also confirm the trend noticed in the main paper regarding the behavior of CAP-UDF-M (LL219-221), which has been often found less accurate than competing methods, but we find that it performs better at higher resolutions.
>
> On 3D Scene, containing complex and larger scale scenes, our iterative scheme is able to provide benefits already at resolution 256, confirming the findings of the main paper.
>
> ## References
> [1] Zhou J. et al., CAP-UDF: Learning Unsigned Distance Functions Progressively From Raw Point Clouds With Consistency-Aware Field Optimization, PAMI, 2024
>
> [2] Q.-Y. Zhou and V. Koltun, “Dense scene reconstruction with points
> of interest,” ACM Transactions on Graphics (TOG), 2013.
>
> [3] Zhou J. et al., Learning Consistency-Aware Unsigned Distance Functions Progressively from Raw Point Clouds, NeurIPS, 2022
>
> [4] Mello Rella E. et al., Neural Vector Fields for Implicit Surface Representation and Inference, IJCV, 2025
>
> [5] Zheng Z. et al., Deep Implicit Templates for 3D Shape Representation, CVPR, 2021
>
> [6] Takikawa T. et al., Variable Bitrate Neural Fields, SIGGRAPH 2022
>
> [7] Nam G. et al., 3D-LDM: Neural Implicit 3D Shape Generation with Latent Diffusion Models, 2022

---

> > ### Comment · Reviewer_YyyE · 2025-08-01
> >
> > I sincerely thank the authors for your extensive reply. To start with, I apologise for the “autoencoder” part. The rebuttal cleared some of my concerns, but I respectfully request further explanations.
> >
> > **Regarding the quality of the UDF used as benchmark**
> >
> > I agree with the authors’ argument that “it is crucial for a meshing method to perform robustly under realistic, imperfect conditions.” However, I am still not convinced that such a bad UDF is still worthy of benchmark for nearly all experiments. Some of the earlier works, for example, NDF, don’t exhibit such severe cases of artefacts (please refer to NDF paper Figure 1). Furthermore, many methods, for example, CAP-UDF-M (extraction part of CAP-UDF), is capable of extracting relatively smooth meshes without post-processing from the UDF learnt by CAP-UDF-L (learning part of CAP-UDF). Please refer to CAP-UDF paper Figure 10.
> >
> > If the authors wish to use a UDF prediction method that is pretrained, I suggest GeoUDF. If the authors wish to highlight their performance under complicated scenes or high-resolution models, I suggest DEUDF. If the authors wish to demonstrate their method under practical use cases, I suggest NeuralUDF. If the authors wish to compare their results under a relatively accurate UDF, I suggest DUDF. Taking considerations of individual characteristics among the heterogeneous UDF learning methods, the current UDF, homogeneous and low-quality, seems insufficient for proving that their proposed method can be used generally and achieve cutting-edge results.
> >
> > **Performance on GT UDF**
> >
> > I noticed that reviewer Ypj2 noted the performance on GT (ground truth) UDF. Speaking of NSDUDF, their teaser image on their official GitHub repo seems to be trained on GT UDF. However, their reported visualisations contain visible holes and artefacts. I wonder if the authors’ proposed method has similar problems with such high-fidelity UDF or even GT UDF. Such artefacts are not prominent in the reported CD (Chamfer distance) provided to reviewer Ypj2. I understand that uploading images or linking to images is not permitted, but there are some other quantitative criteria that reflect the topology of the extracted meshes, for example, the number of holes and the number of connected components.
> >
> > **UDF training strategy**
> >
> > I noticed in supplementary material A.4 that the authors prepared their training data as such: sample ~400K points in the volume. Could you clarify how the distances are determined for these sample points? If the distances are determined by their distances to the ground truth mesh, how does the supervised UDF end up being low-quality?
> >
> > **Regarding the results on CAP-UDF**
> >
> > I thank the authors for providing more results enabling the proposed method on better UDF learning literatures. CAP-UDF is indeed a very classic work, but I still regret not seeing the method tested on more recent UDF learning methods.
> >
> > The authors only compared the performances on two (2) ShapeNet Cars, which contain more than 2000 different models. Models in ShapeNet Cars are heterogeneous: some models have rich details while others are relatively smoother; some models have interior structures, but others don’t. If the authors can test on two models, they should be able to test on several dozen models in a batch.
> >
> > It is also noticeable that the authors reported far better quantitative results on ShapeNet when using CAP-UDF as learning backend, compared to what they reported in Table 2 of the manuscript. This seems to confirm my previous concern with the intrinsic qualities of the UDF learning backend used in the paper. CAP-UDF relies only on point clouds, and it can achieve better performance than the paper. This makes the relatively unfamiliar UDF learning backend more out of the place. The authors provided more examples that also adopted similar architecture. However, they are mostly no later than 2022 with no exception to the IJCV one, which appeared on arXiv as early as April 2022 (2204.06552) and not published until 2025.
> >
> > **I have additional questions about the experimental results**
> >
> > No results on DMUDF are provided, which is perplexing. Using DMUDF is quite straightforward (a pip command is provided) and should be relatively easy to be used with CAP-UDF.
> >
> > This work is built on NSD-UDF. However, for ShapeNet Car and 3D scenes in 128 resultion, the proposed method achieves worse results than NSD-UDF. I politely request that the authors investigate the cause. Does this mean that the proposed method might negatively impact the reconstruction fidelity under some circumstances?
> >
> > It is also noted that the reported performances on CAP-UDF with DCUDF differ dramatically with the provided results by the authors of DCUDF (Table 1). Is there any significant difference with the hyperparameter choice of DCUDF-T? I politely request that the authors clearly state the experimental setup of DCUDF-T.
> >
> > Sorry for bothering you.

---

> > > ### Author Response · Authors · 2025-08-05
> > > **Clarifications**
> > >
> > > We appreciate that some of the earlier misunderstandings have been resolved. It seems to us that the reviewer's remaining concern centers around the quality of the UDF used in our experiments, partially based on the visual appearance of certain examples. This leads the reviewer to  suggest that our UDF learning setup is too weak or outdated to serve as a fair benchmark. We beg to  differ and would like to clarify a few key points before answering specific questions in the reviewer's last comment:
> > >
> > > - Our results are shown at **$512^3$ resolution**, whereas most prior works report visualizations at **$256^3$ or lower**. UDFs that look smooth at lower resolution can reveal noise or gaps when queried more finely. Our goal is to **push the boundary of high-resolution meshing**, as reflected in our title and motivation. Previous methods such as DualMesh-UDF also highlight the challenge of meshing learned UDFs at higher resolutions.
> > > - **Visualization settings** (renderer, brdf, normal smoothing) can significantly affect perceived quality, sometimes masking or exaggerating artifacts. We kindly invite the reviewer to take a closer look at **Figure 10 of CAP-UDF**: although those meshes were extracted at **only $256^3$ resolution**, they still exhibit noticeable blockiness and holes upon close inspection—artifacts that may be less obvious without zooming in, as we did in our figures. This suggests that such issues are not unique to our method or UDF setup, but are part of the broader challenge in high-resolution meshing from learned UDFs.
> > > - **Our experimental setup is consistent with or more comprehensive than other UDF meshing papers, as we show in the table below**. Most prior works **quantitatively** evaluate on 2–3 UDF representations. In contrast, we test across three setups: one auto-decoder baseline (main paper), one higher-quality variant (supplement), and one modern method (CAP-UDF-L, added in rebuttal). Our method performs better than the baselines in all three cases. We provide an additional experiment with CAP-UDF-L below, and we are running a second modern architecture (DiffUDF), which we will add as soon as it finishes. Note on the table below: DCUDF also tests on SDF-to-UDF conversion scenarios, which are not UDF representations, and shows qualitatives on NDF, so we did not include them in the table.
> > > - There are certainly many robust UDF learning methods that could further strengthen the meshing evaluation. However, new challenging scenarios will inevitably arise, such as recovering internal 3D structures from noisy or incomplete data, and the fundamental issue of UDF quality near the surfaces remains unresolved. As such, extracting meshes from imperfect, realistic UDFs remains a highly relevant problem. We believe progress is needed both in UDF learning and in robust surface extraction. Our work targets the latter, and we see it as a timely and necessary contribution given the current UDF landscape.
> > >
> > > | Method         | UDF Backbone(s) Used                                     | Datasets Evaluated                                   | Venue              |
> > > |----------------|----------------------------------------------------------|------------------------------------------------------|--------------------|
> > > | MeshUDF        | Auto-decoder (DeepSDF-style), NDF                        | MGN (auto-dec), ShapeNet Cars (NDF)                  | ECCV 2022          |
> > > | DCUDF          | CAP-UDF-L, MLP                                             | DeepFashion3D (MLP), ShapeNet Cars (CAP-UDF-L)         | SIGGRAPH Asia 2023 |
> > > | DualMesh-UDF   | Overfitting Siren, Softplus MLP, NDF                                   | Thingi10K, MGN, ABC (Siren); ShapeNet Cars (NDF)     | ICCV 2023          |
> > > | NSD-UDF        | GT, Auto-decoder (DeepSDF-style)                         | ABC (GT only), MGN, ShapeNet Cars                    | ECCV 2024          |
> > > | Ours           | ReLU auto-decoder, Softplus auto-decoder, CAP-UDF-L, DiffUDF (in progress)  | MGN, ShapeNet Cars, Chairs, Planes, 3D Scenes (CAP-UDF-L)  | Under Review       |
> > >
> > > We hope this clarifies that our intent has always been to offer a **rigorous, fair, and forward-looking evaluation**—one that is consistent with prior work, and in several respects, extends it. We now respond to the reviewer's specific questions in detail below.

---

> > > > ### Author Response · Authors · 2025-08-05
> > > > **Detailed responses - Part 1**
> > > >
> > > > ## Regarding the quality of the UDF used as benchmark
> > > >
> > > > > "However, I am still not convinced that such a bad UDF is still worthy of a benchmark for nearly all experiments." The authors should consider GeoUDF, DEUDF, NeuralUDF, DUDF.
> > > >
> > > > As mentioned above, we respectfully believe that the UDFs used in our experiments are valid and meaningful for evaluating mesh extraction—particularly under realistic, imperfect conditions, which are common in practice. In response to the reviewer's suggestion, we have also included results using CAP-UDF-L, a more recent and higher-quality UDF method, bringing our total to three distinct backbones. This is in line with, or exceeds, the number tested in most prior meshing papers. We have additionally expanded our CAP-UDF-L results in this response and, if time permits, we will share results using an additional method (e.g., DiffUDF) in the coming days.
> > > >
> > > > More importantly, our core contribution is a learnable algorithm that identifies reliable cues in a UDF field and propagates them robustly in an iterative manner. This paradigm is intentionally general and, in our view, broadly applicable — precisely because learned UDFs are often imperfect, and high-resolution meshing is especially sensitive to noise and ambiguity. Across all experiments presented, we find no indication that our method is tailored to any specific UDF setup. Additionally, the model is fairly lightweight and fast to train (around 2 hours), making it adaptable to new scenarios. While we agree that testing on additional backbones would add breadth, we believe the current setup already demonstrates the algorithm's validity and practical value effectively.
> > > >
> > > > > NDF doesn't exhibit such severe artefacts (see NDF paper Figure 1)
> > > >
> > > > We believe this impression comes primarily from NDF's mesh extraction pipeline, not from the quality of its learned UDF field.
> > > > 1) **NDF's mesh extraction pipeline hides artifacts, but at high cost:**  NDF meshing pipeline, by design, excels at recovering surface even when the UDF does not reach zero. It optimizes a dense point cloud toward field minima via gradient descent, followed by surface reconstruction using the Ball Pivoting Algorithm (BPA). The catch is that BPA can take hours, its parameters are nontrivial to tune, and when misconfigured, it often produces holes. Even then, the resulting meshes are often noisy, because they are triangulated from a dense point cloud.
> > > > 2) **The learned NDF's UDF itself is not particularly clean:** Figure 10 of DCUDF shows that when the UDFs from NDF networks are meshed using faster extractors like CAP-UDF-M or MeshUDF, the results contain clear artifacts. Figures 10, 12, and 13 of the CAP-UDF paper further support this, showing visible holes and blockiness in meshes extracted from NDF UDFs. These examples indicate that the seemingly smooth results in the original NDF paper are a result of its heavy processing pipeline — not of a particularly well-learned UDF field.
> > > > 3) **We show results at resolution 512.** In figure A.5 of our supplementary we show resolution 256 instead, and all methods show less blockiness and more complete surfaces. Previous methods such as DualMesh-UDF also highlight the challenge of meshing at higher resolutions.
> > > >
> > > >
> > > > > CAP-UDF-M can extract relatively smooth meshes (see CAP-UDF Figure 10)
> > > >
> > > >
> > > > Figure 10 of CAP-UDF exhibits a blockiness similar to that of the figures of MC-based methods in our submission. In the top row, for example, it is very visible on and around the rear tire. In all cases, the perceived blockiness depends on the rendering settings. In practice, all methods that rely on MC for the final triangulation will exhibit the same level of blockiness, because they all use the same triangulation and field interpolation rules. This is true for CAP-UDF-M, MeshUDF, NSD-UDF+MC and Ours+MC.
> > > >
> > > >
> > > > ## Performance on GT UDF
> > > >
> > > >
> > > > We thank the reviewer for raising this point. We confirm that our iterative method offers little to no benefit when applied to perfect UDFs. Logically, there is nothing to gain from iterative modeling when the UDF field is already accurate — surface extraction in such cases relies entirely on local information, and most standard meshing methods can recover clean surfaces directly.
> > > >
> > > > Our method is explicitly designed for the opposite setting: to identify and propagate reliable cues in imperfect or ambiguous UDFs, where local information may be noisy, inconsistent, or missing. In the case of ground-truth UDFs, we find that our updates contribute little and can sometimes introduce minor artifacts — which is consistent with the method's intended use, and potentially influenced by its noise augmentations during training.
> > > >
> > > > While testing on ground-truth fields is a valid stress test, we believe it does not reflect the practical challenges our work aims to address.

---

> > > > > ### Author Response · Authors · 2025-08-05
> > > > > **Detailed responses - Part 2**
> > > > >
> > > > > ## UDF training strategy
> > > > > > Could you clarify how the distances are determined for these sample points?
> > > > >
> > > > >
> > > > > As mentioned in the supplementary, we uniformly sample 200K points on the surface and we make two copies of them. To the first copy we add $G(0,\sqrt(0.005))$ noise, to the second we add $G(0,\sqrt(0.0005))$ noise. We only report it here again for clarity. The sampling is carried out after normalization to $[-1,1]^3$. For each sample, we determine it's distance to the normalized GT mesh using the libigl library.
> > > > >
> > > > > > If the distances are determined by their distances to the ground truth mesh, how does the supervised UDF end up being low-quality?
> > > > >
> > > > > Yes. The distances are to the GT mesh but learning this distance field is non-trivial. Like many learned UDFs, they may not reach exactly zero near the surface, but we do not consider this a sign of low quality, rather, it reflects a well-known limitation of many UDF training setups. If the reviewer is referring to a specific failure case, we would be grateful for more details.
> > > > >
> > > > >
> > > > > ## Regarding the results on CAP-UDF
> > > > >
> > > > >
> > > > > > I still regret not seeing the method tested on more recent UDF learning methods
> > > > >
> > > > >
> > > > > As noted earlier, we understand the desire for broader coverage, but testing across many UDF architectures is non-trivial — even more so under the constraints of a rebuttal timeline. To the best of our knowledge, most prior meshing papers evaluate on 2–3 learned UDF setups. In our case, we evaluate on three: two in the main submission (an auto-decoder and a softplus variant) and one additional modern backbone, CAP-UDF-L, included in the rebuttal. This is on par with, or more than, most prior work. We also add a larger experiment on CAP-UDF-L below and we will provide results on DiffUDF as well.
> > > > >
> > > > >
> > > > > > The authors only compared the performances on two (2) ShapeNet Cars
> > > > >
> > > > >
> > > > > To avoid any fairness problems in how we set the experiments, we used the data provided by CAP-UDF, 2 cars and 5 scenes, as is. Since the data is provided with the original work, it presumably produces high quality results.
> > > > >
> > > > >
> > > > > > If the authors can test on two models, they should be able to test on several dozen models in a batch
> > > > >
> > > > >
> > > > > Certainly. Please keep in mind that these results take time, and our time and compute are limited, so the number of shapes can only be limited. Each network takes around 30m for training and some meshing methods, including DCUDF, take a very long time to execute. We managed to run all the MC-based algorithms on 20 shapes, which we report at the end.
> > > > >
> > > > >
> > > > > > The higher quality of CAP-UDF-L "makes the relatively unfamiliar UDF learning backend more out of the place."
> > > > >
> > > > >
> > > > > There is no doubt that the meshes extracted from CAP-UDF-L are higher quality than the ones produced by ReLU-based autodecoders, when used with the same settings. However auto-decoder scenarios are used not because they achieve the highest UDF quality, but because they enable different tasks. CAP-UDF-L is for reconstruction from point clouds. Auto-decoders are useful to learn a smooth latent space of shapes, useful for example in shape optimization scenarios.
> > > > >
> > > > >
> > > > > ## Additional concerns
> > > > >
> > > > > > No results on DMUDF are provided in the rebuttal
> > > > >
> > > > > We clarify that DMUDF is a dual method and is evaluated separately in the dual-methods table in the main paper. In the rebuttal, due to time constraints, we focused only on primal methods and did not include any dual-method baselines — this includes UNDC, DMUDF, NSD-UDF+DMUDF, and Ours+DMUDF, which are typically evaluated together. As shown in the main paper, combining our method with DMUDF improves performance over standalone DMUDF, but we do not identify a clear winner between primal and dual pipelines, as each has strengths and weaknesses depending on the metric. We will be happy to include all dual methods (UNDC, DMUDF, NSD-UDF+DMUDF and Ours+DMUDF) for the new experiments upon acceptance.
> > > > >
> > > > > We also note that DMUDF requires evaluation at $2^k+1$ grid resolutions (e.g., 129, 257, 513), which makes it slightly misaligned with our current MC resolution settings (128, 256, 512). This occurs because primal and dual methods define resolutions differently. While this is a minor issue, it does add overhead.
> > > > >
> > > > >
> > > > > > "For ShapeNet Car and 3D scenes in 128 resolution, the proposed method achieves worse results than NSD-UDF. [...] Does this mean that the proposed method might negatively impact the reconstruction fidelity under some circumstances?"
> > > > >
> > > > > Yes, we think so. Our method is designed to be robust to imperfect or noisy UDFs, particularly at high resolutions where conventional methods often struggle. At lower resolutions (e.g., 128³), the UDF field is often already clean enough that additional refinement provides limited gain — and in some cases, small artifacts may arise due to interpolation or iterative updates. This tradeoff is also acknowledged in our paper, and we believe it reflects a natural complementarity between different meshing strategies.

---

> > > > > > ### Author Response · Authors · 2025-08-05
> > > > > > **Detailed responses - Part 3 and Additional Results**
> > > > > >
> > > > > > > The "performances on CAP-UDF with DCUDF differ dramatically with the provided results by the authors of DCUDF (Table 1)"
> > > > > >
> > > > > > The numerical values are not comparable between our submission and DCUDF. We compute the CD by sampling 2M points on both the reconstructed and the GT meshes. We then compute the first part of the CD by comparing the point cloud from the reconstructed mesh directly to the ground truth **mesh**, and we compute the second part by doing the opposite, thus yielding very precise metrics with a lower bound equal to 0. DCUDF does not specify how they compute their CD or on how many points, to the best of our knowledge. Notice also that the scores reported on their paper are $\times 10^{-3}$ whereas ours are lower for most methods, since they are $\times 10^{-5}$. As to why in their paper the CD scores between existing methods and DCUDF are similar while in our submission they are more different, we believe there might be two potential explanations. The first is that our CD scores are very precise, and could highlight defects and smaller missing regions that were not originally found by the authors. The second is that, on page 7 of their paper on arXiv, right below Table 1, they mention "We generated double-layered meshes". This suggests that they might have not applied the cut part of their algorithm, resulting in a non-thin surface.
> > > > > >
> > > > > >
> > > > > > > Is there any significant difference with the hyperparameter choice of DCUDF-T? I politely request that the authors clearly state the experimental setup of DCUDF-T.
> > > > > >
> > > > > >
> > > > > > We ran the DCUDF code as found in the author's repository, except for the threshold parameter. This parameter depends on both the starting resolution and the mesh, so it is not trivial to set it optimally. In the DCUDF paper they find that a threshold value of 0.0025 is optimal in their testing scenarios and for resolution 256. To get the scores for DCUDF-T, we ran DCUDF with 4 different thresholds at each resolution, and we selected the best ones for each resolution. We believe this tuning to be unfair to other methods that don't require it, especially considering that to tune it we are effectively using the test set, which shouldn't be done for fairness to the other baselines. We report in the table below the CD results of all the tested settings of DCUDF on CAP-UDF-L with the 2 provided ShapeNet Cars. Notice that, in this test, 0.005 seems a better threshold for resolution 256, highlighting the difficulty of executing DCUDF correctly. We also noticed a mistake in the table provided in the rebuttal, for DCUDF at resolution 512 we mistakenly provided the results with threshold 0.005. The best results we could produce at resolution 512, as seen from the table below, were with threshold 0.0025: CD 87.5, F1 72.1, IC 88.7. This does not change any of our conclusions, but we noticed this small error and want to correct it.
> > > > > >
> > > > > >
> > > > > > | CAP-UDF-L 2 ShapeNet Cars | CD ↓ (128) | CD ↓ (256) | CD ↓ (512) |
> > > > > > |---------------------------|------------|------------|------------|
> > > > > > | DCUDF 0.02                | 306        |            |            |
> > > > > > | DCUDF 0.01                | 159        | 129        |            |
> > > > > > | DCUDF 0.005               | 1307       | 86.5       | 112        |
> > > > > > | DCUDF 0.0025              | 7410       | 817        | 87.5       |
> > > > > > | DCUDF 0.001               |            | Failed     | 1180       |
> > > > > > | DCUDF 0.0005              |            |            | Failed     |
> > > > > >
> > > > > >
> > > > > >
> > > > > >
> > > > > > # Additional results on CAP-UDF-L
> > > > > >
> > > > > > Following the suggestion of the reviewer, we trained CAP-UDF-L networks on the first 20 ShapeNet cars, sampling 10k noise-free points on each shape to use as input point clouds and using the code provided by the authors. The following table shows median metrics for MC-based methods, as in the main paper. DCUDF has been run with 4 thresholds per resolution, and the best one has been picked, as above.
> > > > > >
> > > > > >
> > > > > > | CAP-UDF-L 20 ShapeNet cars | CD ↓ (128) | F1 ↑ (128) | IC ↑ (128) | CD ↓ (256) | F1 ↑ (256) | IC ↑ (256) | CD ↓ (512) | F1 ↑ (512) | IC ↑ (512) |
> > > > > > |------------------|------------|------------|------------|------------|------------|------------|------------|------------|------------|
> > > > > > | CAP-UDF-M        | 10.7       | 44.5       | 85.8       | 7.51       | 48.0       | 86.6       | 8.11       | 48.1       | 86.4       |
> > > > > > | MeshUDF          | 9.83       | 45.4       | 85.7       | 8.65       | 47.5       | 86.1       | 8.46       | 47.6       | 85.9       |
> > > > > > | DCUDF-T          | 45.5       | 41.4       | 85.5       | 31.1       | 45.5       | 85.7       | 31.5       | 45.4       | 85.8       |
> > > > > > | NSD-UDF+MC       | 8.14       | 47.1       | 86.4       | 8.76       | 47.9       | 86.3       | 9.85       | 47.3       | 85.9       |
> > > > > > | Ours+MC          | **7.03**       | **48.3**       | **86.8**       | **7.22**       | **50.1**       | **87.1**       | **7.82**       | **49.8**       | **86.6**       |

---

> > > > > > > ### Comment · Reviewer_YyyE · 2025-08-05
> > > > > > >
> > > > > > > Thank you for your detailed rebuttal and for addressing my previous comments. I appreciate the effort you've put into this revision. I still have some concerns, and I'd like to provide some further feedback to help improve the paper.
> > > > > > >
> > > > > > > **The motivation for high-resolution meshing**
> > > > > > >
> > > > > > > While your paper focuses on high-resolution meshing, the current experimental results don't fully justify the need for a grid of size $512^3$ to extract these specific models. The models you've used are largely smooth and don't seem to require such a high resolution to capture their details.
> > > > > > >
> > > > > > > If, as you state, "our goal is to push the boundary of high-resolution meshing," you should use models that genuinely require a high grid resolution to capture intricate details. For example, the DEUDF paper's Figure 2 shows results on models with rich details, but this paper, unfortunately, lacks a similar discussion.
> > > > > > >
> > > > > > > Therefore, to truly demonstrate the benefits of your approach at higher grid resolutions, you should use models with genuinely rich, fine-grained details. The experiments on 3D scenes in CAP-UDF-L are helpful, but they aren't sufficient on their own.
> > > > > > >
> > > > > > > **The unique noise of other UDF methods**
> > > > > > >
> > > > > > > I agree that UDF methods generally suffer from noise, which can lead to holes and blockiness in the extracted meshes. However, the issues in your results are quite unique. The extracted surfaces in this paper often exhibit a noticeable waviness. I haven't seen this specific type of artifact so prominently in any other work. As Reviewer Ypj2 mentioned, the results appear "voxelised." In Figure A.5 of the supplementary material, all methods show less blockiness and more complete surfaces, but the waviness persists. Could you explain the underlying cause of this specific type of noise?
> > > > > > >
> > > > > > > **Does a UDF derived from GT mesh qualify as real-world scenario?**
> > > > > > >
> > > > > > > You state that your work aims to solve the problem of UDF mesh extraction in real-world scenarios. However, a UDF trained from a GT mesh can't be considered truly "realistic." In a real-world setting, you don't have a GT mesh to train your network with. You need a GT mesh to derive a UDF, yet you need a UDF to obtain the mesh, making a chicken-and-egg problem. Instead, you would compute the UDF from a point cloud or from multi-view images. Using a distance field derived from a GT mesh makes it difficult to validate the method's performance in a real-world environment.
> > > > > > >
> > > > > > > **Performance on GT UDF**
> > > > > > >
> > > > > > > I'm concerned that your method performs worse than NSDUDF on the GT UDF. This suggests that in some cases, the iterative approach is detrimental. How do we measure whether the iterative result is a positive improvement? This ties back to Reviewer Ypj2's point about the lack of theoretical support for your method. How can we be sure that the iteration leads to a stable and positive outcome? You mentioned that the degradation is related to the use of noise augmentations, but without seeing the relevant results, I can't confirm whether this is a flaw in the iterative method itself.
> > > > > > > Furthermore, please provide metrics such as the number of holes, which would help reviewers determine if your method, like NSDUDF, also suffers from topological flaws on GT UDF.
> > > > > > >
> > > > > > > **Diversity on methods tested**
> > > > > > >
> > > > > > > I agree that 2–3 backbones are sufficient. However, as I mentioned previously, I don't endorse the current framework, as it has a unique and noticeable noise pattern and requires a GT mesh for training. The results from CAP-UDF-L clearly reflect this issue, where all methods show improved results. Using a better backbone also highlights a potential opportunity for your paper. However, demonstrating your method's superiority on the current backbone, an overly noisy UDF, makes it difficult for readers to distinguish whether the waviness in the results is due to your method or the poor-quality UDF.
> > > > > > >
> > > > > > > **"Chamfer Distance"**
> > > > > > >
> > > > > > > The term "Chamfer distance" is a metric used to quantify the similarity between two sets of points. The way you've calculated it in this paper could cause confusion, and I suggest you use a different name to avoid ambiguity.
> > > > > > >
> > > > > > > **DCUDF**
> > > > > > >
> > > > > > > I have used DCUDF before and agree with its performance, therefore your provided results are quite unconvincing to me. Based on Figure A.5 (missing components) and your remarks I notice that you seem to have applied the mini-cut postprocessing step *against their recommendations in the paper*. This post-processing operation is only suitable for simple manifolds, not for complex, non-manifold ones like ShapeNet Cars. Please report the Chamfer Distance when minicut postprocessing is not used.

---

> > > > > > > > ### Author Response · Authors · 2025-08-09
> > > > > > > > **Discussion 2.1**
> > > > > > > >
> > > > > > > > Thank you for the follow-up. We apologize for the delayed response. Below, we provide additional results and address your questions.
> > > > > > > >
> > > > > > > > # Surface extraction from DiffUDF
> > > > > > > > Following the reviewer's request, we set up the same scenario presented above for CAP-UDF-L and used DiffUDF to learn a UDF. DiffUDF proposes a training scheme and gradient computation strategy that make the UDF differentiable, helping to reduce some of the existing challenges of UDF representation and reducing training convergence time, at the cost of slower inference speed when computing UDF gradients. The following table shows median metrics for MC-based methods, as in the main paper. DCUDF has been run with the same settings found optimal for the CAP-UDF-L experiment on the same shapes.
> > > > > > > >
> > > > > > > > We notice that, once again, we provide the best results at high resolutions (512 and 256), which is the main goal of the proposed method. Note also that the increased accuracy provided by DiffUDF over CAP-UDF-L comes at the cost of meshing speed, because computing gradients on DiffUDF requires second-order derivates, making all methods significantly slower. In our tests most methods are around three times slower, except for DCUDF which, due to its optimization procedure requiring gradient queries at every step, can take several hours per mesh at resolution 512.
> > > > > > > >
> > > > > > > > | DiffUDF 20 ShapeNet cars | CD ↓ (128) | F1 ↑ (128) | IC ↑ (128) | CD ↓ (256) | F1 ↑ (256) | IC ↑ (256) | CD ↓ (512) | F1 ↑ (512) | IC ↑ (512) |
> > > > > > > > |--------------------------|------------|------------|------------|------------|------------|------------|------------|------------|------------|
> > > > > > > > | CAP-UDF-M                | 6.71       | 58.0       | 86.3       | 3.72       | 67.9       | 86.7       | 4.60       | 67.1       | 84.4       |
> > > > > > > > | MeshUDF                  | 9.02       | 59.7       | 86.7       | 5.18       | 65.9       | 87.2       | 3.83       | 67.4       | 86.4       |
> > > > > > > > | DCUDF-T                  | 32.7       | 49.6       | **89.6**       | 25.5       | 58.3       | 86.4       | 17.2       | 60.1       | 84.3       |
> > > > > > > > | NSD-UDF+MC               | **4.18**       | **65.7**       | 87.7       | 3.63       | **70.0**       | 86.9       | 5.63       | 66.0       | 82.0       |
> > > > > > > > | Ours+MC                  | 5.46       | 62.3       | 87.4       | **3.45**       | 68.9       | **88.3**       | **3.03**       | **71.7**       | **88.1**       |
> > > > > > > >
> > > > > > > > ---
> > > > > > > > ---
> > > > > > > >
> > > > > > > > ## 1. Motivation for High-Resolution Meshing
> > > > > > > > > "The models you've used are largely smooth and don't seem to require such a high resolution to capture their details."
> > > > > > > >
> > > > > > > > > "You should use models that genuinely require a high grid resolution to capture intricate details."
> > > > > > > >
> > > > > > > > First, our paper emphasizes the unique challenges of high-resolution meshing - challenges that arise **even for relatively simple shapes**. At finer resolutions, surface extraction depends on highly accurate UDF values near the zero level-set. Even small errors can lead to visible artifacts, and most existing meshing methods degrade significantly under these conditions, regardless of the input shape's complexity.
> > > > > > > >
> > > > > > > > Of course, testing on shapes with extremely fine-grained details could make the benefits of high-resolution meshing more visually obvious - assuming those details are faithfully captured in the UDF field. However, even simple shapes benefit from higher resolution, for example in curved regions. We believe extreme cases are not necessary to evaluate the performance of high-resolution meshing algorithms. As also highlighted in prior works (e.g., DualMesh-UDF), the challenges of meshing at high resolution are well established. To our knowledge, we are the first to systematically evaluate surface extraction at $\gt256^3$ resolution with a dedicated methodology designed specifically for this demanding setting.
> > > > > > > >
> > > > > > > > Second, we believe our experiments already span both simple and complex structures. In addition to garments (MGN), we test on three diverse ShapeNet categories - cars, chairs, and planes - each containing rich geometric features like internal structures, junctions, wheels, and curved surfaces. We also included large-scale 3D scenes (CAP-UDF-L) in the rebuttal, which exhibit further complexity. While we appreciate the suggestion to explore even more detail-rich models (e.g., DEUDF-style), based on our current results, we believe the existing benchmarks already demonstrate the core strengths of our method under high-resolution settings.

---

> > > > > > > > > ### Author Response · Authors · 2025-08-09
> > > > > > > > > **Discussion 2.2**
> > > > > > > > >
> > > > > > > > > ## 2. Waviness and Noise Pattern
> > > > > > > > > > "The extracted surfaces in this paper often exhibit a noticeable waviness... I haven't seen this specific type of artifact so prominently in any other work."
> > > > > > > > >
> > > > > > > > > > "Could you explain the underlying cause of this specific type of noise?"
> > > > > > > > >
> > > > > > > > > As we explained in the previous response, the blockiness/waviness is shared by all MC-based methods and is partly due to the triangulation rules. Dual methods and DCUDF suffer less thanks to their optimization procedures.
> > > > > > > > >
> > > > > > > > > In more detail, we observed that this effect often occurs in surface regions that intersect the grid cells diagonally. In such configurations, some grid vertices lie very close to the surface, where UDF values and gradients tend to be less reliable. This makes triangulation, which assumes perfect linearity of the field, more error-prone, leading to visible artifacts. The issue becomes more pronounced at higher resolutions, where the tri-linear interpolation occurs over finer distances and thus relies on higher precision. While this appears to be a key cause, we remain open to the possibility that other factors may also contribute to this behavior, though a deeper investigation would be outside the scope of this paper.
> > > > > > > > >
> > > > > > > > > This effect, however, is **not unique** to our method or our auto-decoders. We kindly ask the reviewer to look at figure 1 and 10 of DCUDF: MC-based methods such as CAP-UDF-M and MeshUDF exhibit the same blockiness and waviness as in our figures, if not more. This seems to be a frequent issue with UDF representations, which is exacerbated by higher resolutions, as noted also by the reviewer.
> > > > > > > > >
> > > > > > > > > ## 3. Real world scenarios
> > > > > > > > > > "In a real-world setting, you don't have a GT mesh to train your network with." Is it still a "real-world" scenario? Real-world scenarios should involve reconstruction from point clouds instead.
> > > > > > > > >
> > > > > > > > > We agree with the reviewer that, in many real-world scenarios, meshes are reconstructed from partial inputs such as point clouds rather than from GT meshes. Our original experiments followed the standard evaluation protocol in prior UDF meshing work (e.g., MeshUDF, NSD-UDF), where auto-decoder UDFs are trained from GT meshes. In fact, most prior UDF meshing algorithms have been evaluated on UDFs reconstructed from the same meshes the UDF networks were trained on (MeshUDF, DualMesh-UDF, NSD-UDF).
> > > > > > > > >
> > > > > > > > > Following the reviewer's suggestion, in the rebuttal and discussion we have extended our evaluation to more realistic inputs by testing on UDFs generated from point clouds using **CAP-UDF-L** for ShapeNet Cars and 3D scenes (following the official repository), and additional cars, all following the experimental setup in Sections 4.1 and 4.3 of the CAP-UDF paper. We have also run further experiments with **DiffUDF**, as shown above.
> > > > > > > > >
> > > > > > > > > These additions complement our original benchmarks and show that our method is compatible with both auto-decoders trained with GT meshes, as well as UDFs reconstructed from point clouds, consistently improving mesh quality in imperfect-field scenarios.

---

> > > > > > > > > > ### Author Response · Authors · 2025-08-09
> > > > > > > > > > **Discussion 2.3**
> > > > > > > > > >
> > > > > > > > > > ## 4. Performance on GT UDFs and method’s convergence
> > > > > > > > > > > "I'm concerned that your method performs worse than NSDUDF on the GT UDF."
> > > > > > > > > >
> > > > > > > > > > We do not fully resonate with the reviewer's continued insistence on further evaluation using ground-truth UDFs. We believe this test is not meaningful for assessing our method's effectiveness or for drawing theoretical conclusions. Our method is explicitly designed for **imperfect UDFs**, where reliable cues must be identified and propagated through iterative refinement. In a **perfect UDF**, such modeling is unnecessary and offers no benefit — all relevant surface information is already present locally.
> > > > > > > > > >
> > > > > > > > > > Applying our approach in this setting is akin to **running a denoising network on a clean image**: there is nothing to correct, and the process may even introduce slight distortions. Therefore, we respectfully argue that this scenario falls outside the intended use case of our method, and conclusions drawn from it would be misleading.
> > > > > > > > > >
> > > > > > > > > > Nonetheless, we provide here the number of holes, as requested. We computed holes as surface boundaries, similarly to the description provided by DCUDF, for 20 ShapeNet Cars at resolution 512. We report the average.
> > > > > > > > > >
> > > > > > > > > > | GT  | CAP-UDF | MeshUDF | DCUDF-T | DCUDF-T w/o cut | NSD-UDF+MC | Ours+MC |
> > > > > > > > > > |-----|---------|---------|---------|-----------------|------------|---------|
> > > > > > > > > > | 783 | 10024   | 428     | 7       | 2.65            | 8280       | 9284    |
> > > > > > > > > >
> > > > > > > > > > We notice the original meshes have a large number of boundaries, because they contain multiple detailed components and inner structures. None of the methods faithfully recovers the correct mesh topology. Methods that rely directly on MC triangulation, such as CAP-UDF, NSD-UDF+MC and Ours+MC, tend to suffer from micro-holes and gaps between some of the faces. MeshUDF uses a heuristic specifically designed to reduce this behavior and connect as many portions of the surface as possible, explaining the lower number of holes. DCUDF-T starts from an inflated mesh, so it generally tends to have fewer holes. Simple postprocessing steps can be applied to all methods to improve the final mesh quality. We take as an example the first of the ShapeNet Cars (object 100715345ee54d7ae38b52b4ee9d36a3), and we apply trimesh-based postprocessing (fill small holes, merge close vertices, remove spurious faces):
> > > > > > > > > >
> > > > > > > > > > |                        | GT   | CAP-UDF | MeshUDF | DCUDF-T | DCUDF-T w/o cut | NSD-UDF+MC | Ours+MC |
> > > > > > > > > > |------------------------|------|---------|---------|---------|-----------------|------------|---------|
> > > > > > > > > > | Before post-processing | 1552 | 13265   | 997     | 6       | 2               | 8885       | 9808    |
> > > > > > > > > > | After post-processing  | /    | 2125    | 240     | 6       | 2               | 1387       | 1604    |
> > > > > > > > > >
> > > > > > > > > > None of the methods perfectly reconstructs the original mesh, but Ours+MC is the closest in terms of number of holes.
> > > > > > > > > >
> > > > > > > > > > We emphasize that in the GT UDF setting our method performs comparably to others, offering neither benefit nor significant degradation — which is exactly as expected. This scenario is entirely hypothetical and does not arise in practice, as the reviewer themselves acknowledged. In real-world applications, UDFs are learned from partial or noisy data, and it is in those imperfect settings that our method is designed to operate and shows clear advantages.
> > > > > > > > > >
> > > > > > > > > > > "How can we be sure that the iteration leads to a stable and positive outcome?"
> > > > > > > > > >
> > > > > > > > > > We do not have a theoretical guarantee of convergence, a typical problem shared by most neural network-based approaches in broader literature. We believe we have provided abundant evidence that our method works in a multitude of different scenarios and datasets.

---

> > > > > > > > > > > ### Author Response · Authors · 2025-08-09
> > > > > > > > > > > **Discussion 2.4**
> > > > > > > > > > >
> > > > > > > > > > > ## 5. Backbone Quality + Method Distinction
> > > > > > > > > > > > "I don't endorse the current framework, as it has a unique and noticeable noise pattern and requires a GT mesh for training."
> > > > > > > > > > >
> > > > > > > > > > > First, we clarify that many UDF backbones -including our auto-decoders- are trained from GT meshes. Auto-decoders are designed to be trained with GT meshes, and as we described above, we trained and tested them according to existing literature. NDF, a multiple-shape non-autodecoder network, is also trained on GT meshes. Even single-shape networks are often trained and evaluated with a GT mesh, for example in the DualMesh-UDF paper. Auto-decoders are a standard setup in previous works (e.g., MeshUDF, NSD-UDF) that we simply follow, and there is nothing fundamentally unique or problematic about using auto-decoders.
> > > > > > > > > > >
> > > > > > > > > > > Second, the “waviness” or “voxelized” artifacts mentioned by the reviewer are not specific of our backbone. These artifacts are common across UDF networks and especially prominent in high-resolution meshing due to instability in UDF values near the surface. The reviewer seems to base this claim on a few selected visual examples. However, we kindly ask the reviewer to take a closer look at comparable results in prior work, which show exactly the same patterns:
> > > > > > > > > > > - Figure 1 and 10 of DCUDF
> > > > > > > > > > > - Figure 10 (top row, third column) 14 (top row) and 17 (w/o refinement) of CAP-UDF
> > > > > > > > > > >
> > > > > > > > > > > Lastly, we emphasize that we have now tested our method on four different UDF backbones, including CAP-UDF-L and DiffUDF, both of which do not use auto-decoders. This goes beyond what is commonly done in similar literature and demonstrates that our method is not tied to a specific UDF representation.
> > > > > > > > > > >
> > > > > > > > > > > > "Using a better backbone also highlights a potential opportunity for your paper."
> > > > > > > > > > >
> > > > > > > > > > > We are happy to have provided all the following:
> > > > > > > > > > > 1) A better auto-decoder backbone (supplementary)
> > > > > > > > > > > 2) A **different** use case, involving CAP-UDF-L as a backbone
> > > > > > > > > > > 3) An additional backbone, DiffUDF, as requested
> > > > > > > > > > >
> > > > > > > > > > > ## 6. Chamfer Distance Naming
> > > > > > > > > > > > "The way you've calculated it in this paper could cause confusion, and I suggest you use a different name."
> > > > > > > > > > >
> > > > > > > > > > > To avoid ambiguity, we will rename the metric to “Mesh Chamfer Distance” in the final version of the paper. As already noted in Lines 195–196, we compute the metric by sampling 2 million points on each mesh and calculating the distance from point cloud to mesh and vice versa. We chose this setup to ensure high precision and fairness within our experiments, balancing it against computational cost.
> > > > > > > > > > >
> > > > > > > > > > > We also clarify that different papers use varying sampling strategies  often with far fewer points (typically 100–200k) or without specifying details. As stated in an earlier response, this makes absolute CD scores across papers generally non-comparable. However, within our submission, all methods are evaluated under exactly the same setup, so the reported numbers are reliable for relative comparisons.

---

> > > > > > > > > > > > ### Author Response · Authors · 2025-08-09
> > > > > > > > > > > > **Discussion 2.5**
> > > > > > > > > > > >
> > > > > > > > > > > > ## 7. DCUDF Minicut Evaluation
> > > > > > > > > > > > > "You seem to have applied the mini-cut postprocessing step against their recommendations..."
> > > > > > > > > > > >
> > > > > > > > > > > > We would like to clarify that we did not run DCUDF “against the recommendations”. Our evaluation strictly follows what is described in the DCUDF paper itself. According to Algorithm 1 and multiple mentions throughout the paper — including the abstract — DCUDF attempts to apply the cut, and if it fails due to non-manifoldness, it automatically returns a double-layered surface. This fallback is explicitly documented as the intended behavior of the method.
> > > > > > > > > > > >
> > > > > > > > > > > > Their code repository indicates to disable the cut “for complex models or non-manifold models such as car, scenes, etc”. But this is not indicated in the paper:
> > > > > > > > > > > > 1) It is unclear whether it is for speed purposes or because the method fails otherwise. If it is for the latter reason, it is unclear why. In the paper it is stated that the method returns double layered surfaces when it fails to cut, and we did notice this behavior, so there should not be any need to manually disable the cut.
> > > > > > > > > > > > 2) In a realistic scenario, one does not know in advance whether a UDF or a point cloud will produce a manifold surface or not, so it is unclear when to disable it or not. For example the same car model can be manifold or not, depending on how it is represented. Letting the algorithm decide seems the best option.
> > > > > > > > > > > > 3) Double layered surfaces have an inner volume (i.e. they are not thin). It is up for debate whether they can be considered a desirable meshing outcome for UDFs, since thin surfaces are usually the use case for UDF. Such debate is out of the scope of our paper. Instead, we compared all methods in the same conditions, i.e. extracting thin surfaces, and as described in their corresponding papers.
> > > > > > > > > > > > 4) To be as fair as possible to DCUDF we ran it with multiple threshold configurations, as shown in the previous response, which is already unfair to the other baselines that can be simply run as they are. DCUDF has its advantages, and we state them in our manuscript, but it is notably slower and more difficult to use than the other methods.
> > > > > > > > > > > > 5) Table A.6 of NSD-UDF compares DCUDF with and without the cut part. The CD metrics improve most of the time by avoiding the cut, **but not always**, and they do not achieve better scores than NSD-UDF. We expect similar results here.
> > > > > > > > > > > >
> > > > > > > > > > > > > "Please report the Chamfer Distance when minicut postprocessing is not used."
> > > > > > > > > > > >
> > > > > > > > > > > > We refer to the answer above. However, we decided to run DCUDF without the cut part, as suggested by the reviewer. We ran it on auto-decoder cars, CAP-UDF-L cars and DiffUDF cars, since “car” is one of the reported shapes in which the repository of DCUDF suggests to disable the cut, and since the reviewer explicitly mentions it. We used the same thresholds which we found optimal in the corresponding experiments. We report median scores below.
> > > > > > > > > > > >
> > > > > > > > > > > > | DCUDF w/o cut     | CD ↓ (128) | F1 ↑ (128) | IC ↑ (128) | CD ↓ (256) | F1 ↑ (256) | IC ↑ (256) | CD ↓ (512) | F1 ↑ (512) | IC ↑ (512) |
> > > > > > > > > > > > |-------------------|------------|------------|------------|------------|------------|------------|------------|------------|------------|
> > > > > > > > > > > > | ReLU auto-decoder | 11.4       | 61.7       | 89.4       | 52.0       | 58.9       | 82.3       | 43.1       | 60.4       | 85.6       |
> > > > > > > > > > > > | CAP-UDF-L         | 8.47       | 47.2       | 88.3       | 9.58       | 47.9       | 86.0       | 11.8       | 46.3       | 84.8       |
> > > > > > > > > > > > | DiffUDF           | 12.5       | 56.9       | 90.2       | 5.56       | 69.5       | 87.7       | 5.53       | 69.0       | 87.1       |
> > > > > > > > > > > >
> > > > > > > > > > > > We further highlight that we do not believe these scores are directly comparable to the ones from the tables in the paper or the tables above because they come from thick surfaces, not thin like all the other methods. Even so, by comparing these scores with the table above and with the results in the main paper, the relative ranking of the meshing methods we tested remains the same in most scenarios, with the few exceptions being at low resolution (128), thus the same conclusions apply.
> > > > > > > > > > > >
> > > > > > > > > > > > We thank the reviewer again for the discussion and for the opportunity to clarify our method and contributions, as well as to extend the experimental scope.

---

### Official Review · Reviewer_EmVW · 2025-06-28

**Clarity:** 3
**Significance:** 3
**Originality:** 3
**Rating:** 4
**Confidence:** 4

**Summary:**

This paper presents a new approach for meshing neural Unsigned Distance Fields (UDFs) at high resolution, which are typically more difficult to triangulate than Signed Distance Fields due to the lack of sign changes near the surface. The proposed method introduces an iterative refinement strategy: instead of performing surface extraction in a single pass, the network operates over multiple iterations, progressively improving local predictions by integrating surface information from neighboring voxels. At each iteration, the network predicts pseudo-sign configurations for the vertices of a voxel using UDF values, gradients, and surrounding context. This iterative propagation scheme enables the recovery of finer surface details, closes holes, and stabilizes extraction in noisy or ambiguous regions. The approach is tested across diverse 3D datasets and resolutions, demonstrating improved robustness over existing methods.

**Questions:**

see weaknesses

**Ethical Concerns:**

["NO or VERY MINOR ethics concerns only"]

**Limitations:**

yes

**Paper Formatting Concerns:**

I did not notice any major formatting issues. The paper appears to follow the NeurIPS 2025 formatting guidelines.

**Quality:**

3

**Strengths And Weaknesses:**

The proposed method tackles a well-known challenge in high-resolution UDF meshing by introducing a conceptually simple yet powerful idea: iterative correction based on spatial propagation. This naturally complements the limitations of single-pass algorithms and effectively exploits structure within the field. The design of the network is well-explained and integrates neighboring context without needing complicated geometry-specific heuristics. Additionally, the method is robust to noise and generalizes across multiple datasets and resolutions, showcasing the practicality of the approach for real-world high-resolution meshing tasks.
Several points, however, could be clarified further. First, the approach currently uses a fixed maximum number of iterations (up to 6), selected empirically. Could the authors provide insights into whether an adaptive stopping criterion might be beneficial, or whether performance saturates early? Second, while the filtering strategy for removing high-UDF cells is described, it's unclear whether the model remains robust under different UDF scales or UDFs learned from different network architectures—some discussion on generalizability across UDF sources would be helpful.Third, some related work such as “Learning a More Continuous Zero Level Set in Unsigned Distance Fields through Level Set Projection” could be cited for completeness.

---

> ### Author Rebuttal · Authors · 2025-07-31
>
> > **Q1**: Why is a fixed number of iterations used, and could an adaptive stopping criterion improve the method?
>
> **A1**: Thank you for raising this interesting point. We used a fixed maximum of 6 iterations for simplicity and ease of training, and found that performance tends to saturate after 5–6 passes (see Table 4 and A.3). That said, an adaptive stopping criterion is an interesting direction. We observe that simpler shapes indeed do not require full cycle and can be stopped early to save computation. This can be seen in Table 4, where cars, chairs and planes always benefit from additional iterations, whereas garments, due to their simplicity, only benefit at high resolution. It remains an interesting direction for future work. We also experimented with a per-cell stopping criterion based on output confidence scores, but it did not improve the results.
>
> ***
>
> > **Q2**: Is the model robust to UDFs with different value scales or learned from different architectures? Some discussion on generalizability across UDF sources would be helpful.
>
> **A2**: This is a great question. Our current approach includes resolution normalization of the UDF values (Sec. 3.1), which helps maintain robustness across scales, and assumes that the shapes lie in a $[-1,1]^3$ cube. It can be used also with non-normalized (or differently normalized) shapes by re-normalizing the UDF values after grid sampling, which is what we did in the experiments with the UDF training pipeline of CAP-UDF [1] (see below), since it is not trained on a $[-1,1]^3$ cube. Our method is designed to operate on any pre-trained UDF and does not depend on the specific architecture or training objective used. In fact, we only trained a single network on 80 shapes of the ABC dataset and tested this network on various datasets including two auto-decoders architectures (ReLU-based, in the main paper, and Softplus-based, in the supplementary) trained on MGN, ShapeNet cars, chairs and planes, as well as CAP-UDF [1] networks trained on ShapeNet cars and 3D Scene point clouds in the new experiment shown below. However, we recognize that UDF quality and noise levels can vary across sources and architectures, and we will include a discussion of this in the final version to acknowledge the scope and limitations of the current evaluation.
>
> ***
>
> > **Q3**: Some related work such as “Learning a More Continuous Zero Level Set in Unsigned Distance Fields through Level Set Projection” could be cited for completeness.
>
> **A3**: We would be happy to. Thank you for the suggestion.
>
> ***
>
> ## Surface extraction from a different UDF architecture
>
> *Note: CAP-UDF [1,3] defines a UDF architecture and training as well as a meshing algorithm. To avoid confusion, we refer here to the learning part as CAP-UDF-L, and to the meshing part as CAP-UDF-M, which has been referred to as CAP-UDF in the manuscript.*
>
> We train CAP-UDF-L [1] networks following the official code repository on the point clouds provided by the authors: 2 cars from ShapeNet, which are not part of our other experiments, using 10k points as described in [1]; and 5 scenes from 3D Scene [2], using the 100 $points/m^2$ scenario described in [1]. These networks are trained differently than the ones used in our other experiments, have a different normalization, and are not auto-decoders, providing insights about the performance of our pipeline in a very different scenario.
>
> We found that Copyroom and Lounge from 3D Scene produced very bad metrics across all methods due to a seemingly inconsistent normalization, skewing the means, so we excluded them from the table below. We compare Marching Cubes-based methods using the same metrics as the main paper. CD is $\times 10^{-5}$. Since the number of shapes/scenes is low, we report mean metrics instead of median.
>
> On cars, which are learned using a relatively dense input point cloud, most methods perform relatively well at resolution 128 and 256, with NSD-UDF and our own pipeline closely competing. At 512 a similar scenario as in the main paper arises, with the competing methods sometimes losing accuracy compared to their scores at 256. We also confirm the trend noticed in the main paper regarding the behavior of CAP-UDF-M (LL219-221), which has been often found less accurate than competing methods, but we find that it performs better at higher resolutions.
>
>
> | ShapeNet cars | CD ↓ (128) | F1 ↑ (128) | IC ↑ (128) | CD ↓ (256) | F1 ↑ (256) | IC ↑ (256) | CD ↓ (512) | F1 ↑ (512) | IC ↑ (512) |
> |---------------|------------|------------|------------|------------|------------|------------|------------|------------|------------|
> | CAP-UDF-M       | 7.67       | 66.0       | 88.9       | 2.05       | 74.9       | 90.5       | 1.93       | 76.0       | 90.5       |
> | MeshUDF       | 3.77       | 70.5       | 89.3       | 2.34       | 74.8       | 90.3       | 2.25       | 75.7       | 90.3       |
> | DCUDF-T       | 159        | 58.6       | 85.8       | 86.5       | 71.4       | **90.7**       | 112        | 70.3       | 88.6       |
> | NSD-UDF+MC    | **2.34**       | **72.8**       | **90.0**       | 2.05       | **75.8**       | 90.5       | 2.16       | 75.9       | 90.4       |
> | Ours+MC       | 2.43       | 71.3       | 89.8       | **1.86**       | 75.5       | 90.6       | **1.85**       | **76.3**       | **90.7**       |
>
> On 3D Scene, containing complex and larger scale scenes, our iterative scheme is able to provide benefits already at resolution 256, confirming the findinds of the main paper.
>
>
> | 3D Scene - 100 | CD ↓ (128) | F1 ↑ (128) | IC ↑ (128) | CD ↓ (256) | F1 ↑ (256) | IC ↑ (256) | CD ↓ (512) | F1 ↑ (512) | IC ↑ (512) |
> |----------------|------------|------------|------------|------------|------------|------------|------------|------------|------------|
> | CAP-UDF-M        | 4.27       | 68.0       | 83.8       | 3.65       | 70.1       | 86.1       | 3.57       | 70.5       | 86.1       |
> | MeshUDF        | 4.40       | 68.2       | 84.5       | 3.42       | 69.6       | 86.3       | 4.49       | 69.9       | 84.5       |
> | DCUDF-T        | 279        | 59.7       | 84.0       | 4.75       | 69.9       | 84.6       | 140        | 68.9       | 83.3       |
> | NSD-UDF+MC     | **3.34**       | **69.8**       | **86.7**       | 3.30       | 70.3       | 86.3       | 3.72       | 70.0       | 84.0       |
> | Ours+MC        | 3.44       | 69.3       | 85.3       | **3.07**       | **71.4**       | **86.6**       | **3.08**       | **71.8**       | **86.8**       |
>
>
> ## References
> [1] Zhou J. et al., CAP-UDF: Learning Unsigned Distance Functions Progressively From Raw Point Clouds With Consistency-Aware Field Optimization, PAMI, 2024
>
> [2] Q.-Y. Zhou and V. Koltun, “Dense scene reconstruction with points
> of interest,” ACM Transactions on Graphics (TOG), 2013.
>
> [3] Zhou J. et al., Learning Consistency-Aware Unsigned Distance Functions Progressively from Raw Point Clouds, NeurIPS, 2022

---

> > ### Comment · Reviewer_EmVW · 2025-08-08
> >
> > After reading the authors’ response and considering other reviewers’ comments, I tend to maintain my original score. The authors’ clarifications have adequately addressed my concerns, and I recommend acceptance of the paper.

---

### Official Review · Reviewer_YYpx · 2025-07-01

**Clarity:** 4
**Significance:** 3
**Originality:** 4
**Rating:** 5
**Confidence:** 3

**Summary:**

The paper introduces an iterative scheme to compute pseudo-signs for Unsigned Distance Fields (UDF) mesh extraction. The core idea is to progressively refine the result by taking into consideration the previous interaction pseudo-signs, UDF values and gradients.

**Questions:**

No additional questions.

**Ethical Concerns:**

["NO or VERY MINOR ethics concerns only"]

**Final Justification:**

I acknowledge reading the rebuttal message and reflecting about the points raised by the other reviewers. I decide to maintain my rating, the paper was an interesting reading and deserves publication in my opinion.

**Limitations:**

Yes, they are described in Section 5.

**Paper Formatting Concerns:**

No isuues.

**Quality:**

4

**Strengths And Weaknesses:**

To be honest, I don't see noticeable flaws in the paper, so I will focus on the strengths.

## Strengths

* Excellent motivation. The problem and the reasons why it exists are clearly described. The claims are properly stated and are sound.

* Contextualization is done well. The differences in comparison with related work are discussed in detail. The authors present confident knowledge in the neural UDF literature.

* Good writing. The paper reads well and is easy to follow. The math notation is also good.

* Experiments provide an excellent overview of how the area deals with mesh extraction. The results are adequate for the claims: a method designed to work well on high resolutions, avoiding the noise generated by the difficult of achieving true zeros at the trained UDF.

* About the method itself, its design makes sense. The iterative nature is aligned with the idea of considering the current neighborhood while computing the pseudo-signs. This seems to be a good alternative to analytic approaches, such as computing curvature, for example, which would also provide neighborhood analysis but would be influenced by gradient existence on discontinuities at the surface. I also liked the efforts into making it faster and more robust to noise. Being an iterative procedure, ensuring it has comparable running times to other SOTA methods is important for it to be considered as a viable choice. Noise robustness is also very important because of the focus on high-resolution reconstruction. The worries of the authors are on-point with the necessities of the problem at hand.

## Current assessment

I am not completely familiar with all the current neural UDF literature. I have reviewed a few papers on recent conferences in the subject, but I am more familiar with SDFs. It is possible that my rating decreases in case reasonable important flaws are pointed by the other reviewers. If that is not the case, I recommend acceptance.

---

> ### Author Rebuttal · Authors · 2025-07-31
>
> We sincerely thank the reviewer for the thoughtful and encouraging feedback. We are especially grateful for the comments highlighting the clarity of the writing, the relevance of the motivation, and the soundness of the method design.
>
> > “Excellent motivation... confident knowledge in the neural UDF literature... good writing... results are adequate for the claims”
>
> We appreciate these remarks, and we are glad the motivation and problem setup came across clearly. It was important for us to position the work carefully within the existing literature and articulate why high-resolution UDF meshing remains a challenging and underexplored problem.
>
> > “The iterative nature is aligned with the idea of considering the current neighborhood... a good alternative to analytic approaches.”
>
> Thank you for this insight. We agree that iterative refinement offers a practical and robust alternative to curvature-based or analytic neighborhood methods, especially when gradients may be unreliable near surface discontinuities. We are also glad the design considerations around runtime and noise robustness resonated — these were key goals in making the method not only effective but also practical.
>
> > “I am not completely familiar with all the current neural UDF literature”, my rating may adjust based on other reviews.
>
> We completely understand, and we appreciate your openness. We hope the additional responses and clarifications across reviews help to confirm the soundness of the contribution. We are of course receptive to any further concerns that may be raised and we will gladly address them in revisions.
>
> Thank you again for the constructive and thoughtful review.

---

> > ### Comment · Reviewer_YYpx · 2025-08-01
> >
> > I acknowledge reading the rebuttal message and reflecting about the points raised by the other reviewers. I decide to maintain my rating, the paper was an interesting reading and deserves publication in my opinion.

---

### Official Review · Reviewer_Ypj2 · 2025-07-02

**Clarity:** 3
**Significance:** 3
**Originality:** 3
**Rating:** 4
**Confidence:** 4

**Summary:**

This paper addresses the problem of mesh extraction from neural unsigned distance fields (UDFs). Its main contribution is an iterative technique that leverages neighborhood information from the marching cubes grid to refine surface extraction through multiple passes of a neural network. The approach is shown to perform well on complex and high-resolution shapes.

The method builds upon NSD-UDF, with significant overlap in the methodology section; which would benefit from further polishing. This raises the question of how incremental the contribution is, especially since the main novelty (iterative refinement scheme) is introduced without any theoretical justification. This should be clearly acknowledged in the introduction.

**Questions:**

[Figure 5] Why do the reconstructions appear voxelized? Please clarify.

Since the proposed method does not depend on the UDF being represented by an INR,  how does it perform when applied to the ground-truth UDF?

Most of the reconstructed surfaces shown are low in detail. Can the method be applied to more detailed models, such as the Thai Statue and the Asian Dragon?
Note that these cases would require marching cubes at a resolution higher than 512×512.

Regarding the augmentation term. It appears to be a design choice inherited from NSD-UDF, which showed that this factor improves reconstruction quality. However, given the introduction of a refinement step in the current method, I am curious whether this term is still necessary or beneficial.
Finally, I was also curious about the dataset split. Did the authors divide the data into training and validation sets and report results only on the validation set? If so, could they clarify how this split was performed?

**Ethical Concerns:**

["NO or VERY MINOR ethics concerns only"]

**Final Justification:**

I have carefully reviewed the rebuttal, and the authors addressed most of my comments. Therefore, I will maintain my positive score.
If the paper is accepted for publication, I strongly recommend that the additional experiments, important comments, and missing references be incorporated into the paper and supplementary material.

**Limitations:**

Yes.

**Paper Formatting Concerns:**

No issues.

**Quality:**

3

**Strengths And Weaknesses:**

Strengths:

The claim in [L45] that meshing models should incorporate information from previously extracted surface elements is compelling. The evaluation confirms that the proposed iterative refinement improves reconstruction quality. However, the paper lacks a theoretical explanation for why this approach works.

The authors conduct extensive experiments demonstrating improved reconstructions compared to the state of the art.

Weakness:

[L21] The paper cites DeepSDF as a primary motivation for using SDFs in INRs. However, a more accurate motivation involves the use of the Eikonal equation for regularization. Missing references include:
- Sitzmann et al., Implicit Neural Representations with Periodic Activation Functions, NeurIPS
- Gropp et al., Implicit Geometric Regularization for Learning Shapes, ICML
- Novello et al., Exploring Differential Geometry in Neural Implicits, Computers & Graphics
- Fainstein et al., DUDF: Differentiable Unsigned Distance Fields with Hyperbolic Scaling, CVPR

The sentences in [L28–30] and [L33–34] appear redundant and should be revised for clarity.

[L35] “worsens the problem, as shown in the results section” — Please indicate the exact section number or figure that supports this claim.

[Figure 1] This figure needs refinement. Consider showing the full 3D model with zoom-ins to improve clarity. Also, please include the ground-truth mesh for comparison.

[Section 3.2] This section appears to be mostly implementation details.

---

> ### Author Rebuttal · Authors · 2025-07-31
>
> > **Q1**: Iterative refinement scheme is introduced without any theoretical justification.
>
> **A1**: We agree that a theoretical foundation would strengthen the work. However, providing a mathematical proof for the convergence of the iterative refinement over noisy neural fields is extremely challenging, as it involves complex interactions between spatial context, prediction dynamics, and network behavior - none of which is fully understood in existing literature, to the best of our knowledge. For example [8] compares it to denoising autoencoders, this would be an interesting direction. However, our approach is grounded in prior work where similar iterative strategies have proven effective in structured prediction tasks. We hope the strong empirical results (Table 2, Figure 3 and 4) help support its value in this setting.
>
> ***
>
> > **Q2**: Contribution over NSD-UDF?
>
> **A2**: While our method builds on NSD-UDF, we believe the shift from a single-pass to an iterative, context-aware refinement process is a meaningful advancement, particularly at high resolutions. As shown in Figure 1 and 3 and Table 2, the iterative passes help recover surfaces that NSD-UDF misses. That being said, we will make sure to more clearly highlight where our method extends NSD-UDF and where it introduces new ideas.
>
> ***
>
> > **Q3**: The paper cites DeepSDF as a primary motivation for using SDFs in INRs. However, a more accurate motivation involves the use of the Eikonal equation for regularization with missing citations.
>
> **A3**: Thank you for the suggestion and the papers which we will happily cite. However, we would like to clarify that our contribution is focused on meshing from already trained UDFs, not on learning or regularizing them. As such, the Eikonal equation - which is typically used during UDF training - is not directly relevant to our method. While DeepSDF and related works provide context for using distance fields in INRs, our approach is agnostic to the training objective and instead addresses the challenge of surface extraction at inference time. Nevertheless, we provide results on a network trained with Eikonal loss in section A.5 of the supplementary.
>
> ***
>
> > **Q4**: Redundancy in [L28–30] and [L33–34].
>
> **A4**: We agree. L28-30 should be removed.
>
> ***
>
> > **Q5**: [L35] “worsens the problem, as shown in the results section” — Please indicate the exact section number or figure that supports this claim.
>
> **A5**: This is particularly visible in Figure 3 and A.1, in which the extraction of NSD-UDF gets worse with increasing resolution. A similar phenomenon happens to most of the other methods as well, and can be seen by comparing the same shape across different resolutions in figures A.2 to A.6. We will clarify this in the manuscript.
>
> ***
>
> > **Q6**: [Figure 1] This figure needs refinement. Consider showing the full 3D model with zoom-ins to improve clarity. Also, please include the ground-truth mesh for comparison.
>
> **A6**: Thank you for the suggestion. We agree that including the full shape as well as the ground truth would improve the figure. We will update the paper accordingly.
>
> ***
>
> > **Q7**: [Figure 5] Why do the reconstructions appear voxelized? Please clarify.
>
> **A7**: We purposefully did not apply any post-processing or smoothing algorithm, and we rendered the shapes in Blender using non-smoothed normals, which contribute to making the shapes appear voxelized. We did this to faithfully show the output of each meshing algorithm. This could be mitigated with post-processing and using different rendering techniques. The only naturally non-voxelized extraction method, in our testing, has been DCUDF, thanks to its optimization procedure.
>
> ***
>
> > **Q8**: Since the proposed method does not depend on the UDF being represented by an INR, how does it perform when applied to the ground-truth UDF?
>
> **A8**: That is a good point. We did test the method with clean UDFs, and it works well in that setting. However, since there is no neural noise in that case, the iterative refinement offers little to none additional benefit. We include here the median results on ground-truth ShapeNet cars, comparing the first itearation with the last one, and we will clarify this in the revision. CD is $\times 10^{-5}$.
>
> | GT ShapeNet Cars | CD ↓ (128) | CD ↓ (256) | CD ↓ (512) |
> |------------------|------------|------------|------------|
> | Ours iteration 1        | 1.36       | 0.244      | 0.0294     |
> | Ours iteration 6        | 1.40       | 0.231      | 0.0300     |
>
> ***
>
> > **Q9**: Most of the reconstructed surfaces shown are low in detail. Can the method be applied to more detailed models?
>
> **A9**: We believe the perceived low detail is mostly due to the choice of examples, which may not have fully showcased the complexity present in the dataset. In fact, categories like ShapeNet cars and chairs do contain highly detailed structures, and we will include more representative examples to better highlight our method's performance. Additionally, we have extended our evaluation to more complex cases from the 3D Scene dataset, trained with the learning part of CAP-UDF in order to encode more details than in the scenarios shown in the paper, with quantitative results shown at the end of the rebuttal for reviewer EmVW, to further demonstrate the method's applicability to high-detail models.
>
> ***
>
> > **Q10**: Regarding the augmentation term. It appears to be a design choice inherited from NSD-UDF, which showed that this factor improves reconstruction quality. However, given the introduction of a refinement step in the current method, I am curious whether this term is still necessary or beneficial.
>
> **A10**: Thank you for the question. Yes, we found it remains important. The augmentation (Eq. 4) helps the network learn robustness to noisy gradients and values, which are still present in early iterations. Removing it led to slower convergence and worse performance in our tests, as shown below. We appreciate the opportunity to clarify this design choice and we will add this ablation to the supplementary. CD is $\times 10^{-5}$.
>
> | Auto-dec ShapeNet Cars    | CD ↓ (128) | CD ↓ (256) | CD ↓ (512) |
> |---------------------------|------------|------------|------------|
> | Ours+MC (as in paper)     | 5.64       | 5.23       | 8.84       |
> | Ours+MC (no augmentation) | 8.01       | 12.5       | 47.8       |
>
> ***
>
> > **Q11**: Finally, I was also curious about the dataset split. Did the authors divide the data into training and validation sets and report results only on the validation set? If so, could they clarify how this split was performed?
>
> **A11**: The pipeline has been trained on watertight ABC shapes, and has been tested exclusively on open shapes that belong to different datasets, to ensure that there is no overlap between training and testing.
>
>
> ## References
> [8] Durasov N. et al., Enabling Uncertainty Estimation in Iterative Neural Networks, ICML, 2024

---

> > ### Comment · Reviewer_Ypj2 · 2025-08-04
> >
> > I thank the authors for their efforts in addressing the questions and conducting additional experiments. I agree with Reviewer YyyE on the importance of considering additional metrics to evaluate GT UDF. As also noted by YyyE, I believe that a comparison with DualMesh-UDF [1] should be included in the final version, particularly since it reports results on higher-resolution surfaces.
> >
> > [1] Zhang et al. “Surface Extraction from Neural Unsigned Distance Fields.” ICCV, 2023

---

> > > ### Author Response · Authors · 2025-08-05
> > >
> > > Thank you for the comment. Regarding the GT UDF experiment, we confirm that our method offers little to no benefit in this setting. By design, it targets imperfect UDFs where local cues are noisy or ambiguous. In contrast, ground-truth fields already define the surface clearly at the local level, so iterative refinement adds little and can sometimes introduce artifacts.
> > >
> > > As for DualMesh-UDF, we refer the reviewer to Table 2 in the main paper, where we compare our method to dual-based approaches at higher resolutions. Combining our surface localization model with DualMesh-UDF leads to significant improvements across all metrics on three challenging ShapeNet datasets compared to standalone DualMesh-UDF. However we did not find a clear winner between primal and dual methods. We did not include any of the dual methods (UNDC, DMUDF, NSD-UDF+DMUDF, Ours+DMUDF) for the new experiments in the rebuttal due to time constraints, but we will be happy to include them upon acceptance.
> > >
> > > For both points, we also refer the reviewer to our response to reviewer YyyE for further information.
> > >
> > > We’re happy to provide further clarification if needed.

---

> > > > ### Author Response · Authors · 2025-08-09
> > > >
> > > > We provided a few more detailed answers to reviewer YyyE regarding questions that reviewer Ypj2 partially shared. We report a summary here, more details are in the comments to reviewer YyyE (comments "Detailed responses - Part 1" and sections 2, 4 and 5 of the latest batch of comments).
> > > >
> > > > ## Reason for voxelized figures
> > > > We found that meshes extracted from UDFs are often voxelized, more so at higher resolutions, and we hypothesize that it has to do with the assumption of linearity of the field which characterizes Marching Cubes, an assumption that is increasingly violated by grid points on learned UDFs at higher resolutions. This behavior can become more apparent with certain rendering settings and with highly zoomed-in shapes (like in our figure 5), but it is common in existing works and UDF representations other than our auto-decoders, for example NDFs and MLPs. Figure 1 and 10 of the DCUDF paper show clear examples, as well as figures 10, 14 and 17 of the CAP-UDF paper. Post-processing can be applied to smooth the results significantly.
> > > >
> > > > ## Number of holes on GT meshes
> > > > Our method is not designed for GT meshes, so we believe that an experiment in this setting does not provide insights regarding its behavior in the scenarios it is designed for (imperfect UDFs at high resolutions). Nonetheless, we report the number of holes of MC-based methods along with other considerations in section 4 of the last batch of comments to reviewer YyyE.

---

> > > > > ### Comment · Reviewer_Ypj2 · 2025-08-09
> > > > >
> > > > > Thank you to the authors for providing the additional comments. I will take these into consideration during the reviewer–AC discussion.

---

### Note · Authors · 2025-08-13

We sincerely thank all reviewers for their thoughtful feedback.

We believe the community will benefit from our paper for the following reasons:

1. **Important, underexplored problem:** We tackle high-resolution UDF meshing - a challenging setting where most existing methods degrade - and provide the first systematic evaluation at 512³ resolution across multiple datasets.
2. **Novel, practical approach:** We propose an iterative modeling method that identifies reliable cues in imperfect UDFs and propagates them to refine surfaces, end-to-end. The method is effective, easy to train, simple to integrate, and we will release all source code.
3. **State-of-the-art performance:** We outperform the current SOTA on four diverse datasets, using multiple backbones and both mesh-trained and point-cloud–derived UDFs.

During the rebuttal period, many valuable points were raised, and we worked hard to address them thoroughly. To name a few, we added new experiments (DiffUDF, CAP-UDF-L under multiple settings), clarified key claims, and extended evaluation to more realistic point-cloud scenarios. We thank all four reviewers for their time, effort, and constructive engagement. The discussion has been both encouraging and inspiring, and it has helped us significantly improve the paper.

To sum up, we are glad to have had the opportunity to present and discuss our work. We believe it makes a timely and useful contribution to the field by highlighting the challenges of high-resolution UDF meshing, introducing a novel iterative modeling approach, and setting new state-of-the-art results across multiple datasets. We hope the reviewers, AC, and PC will recognize its novelty, clarity, and practical value for the community.

---

### Decision · Program_Chairs · 2025-09-17

**Decision:**

Accept (poster)

**Comment:**

Strengths
* Well-motivated problem of improving the meshing of learned UDFs with inaccuracies
* Iterative refinement is a simple and sensible approach, as the authors state, "meshing models should incorporate information from previously extracted surface elements" (L45)
  * Specifically, the method builds on top of NSD-UDF for iterative, context-aware refinement

Weaknesses
* Could be viewed as an incremental extension to NSD-UDF (debatable, I do not think this is an issue)
* Some missing experiments (see below)

Under the condition that the authors will incorporate the discussions with the reviewers in the final revision, I would like to recommend accepting this paper. In particular, YyyE requested
* Demonstrate qualitative visualisations on GT UDF
  * Ypj2 also raised questions about applying the method to the GT UDF. While the authors stress that the method is meant to recover mesh on noisy UDFs, experiments on GT UDFs will help clarify the scope of the paper and verify that the method does not introduce new artifacts
* Complete the experiments with DiffUDF and CAP-UDF backbones, and incorporate the qualitative & quantitative results to the main text.
* Demonstrate the results of DCUDF without mini-cut, and incorporate the qualitative visualisations to the main text.

YyyE's suggestion on evaluating with watertight meshes is a good test of the topological consistency.